# NOTE: Robust Continual Test-time Adaptation Against Temporal Correlation

**Taesik Gong, Jongheon Jeong, Taewon Kim, Yewon Kim, Jinwoo Shin, and Sung-Ju Lee**
KAIST
Daejeon, South Korea
{taesik.gong,jongheonj,maxkim139,yewon.e.kim,jinwoos,profsj}@kaist.ac.kr

## Abstract

Test-time adaptation (TTA) is an emerging paradigm that addresses distributional shifts between training and testing phases without additional data acquisition or labeling cost; only unlabeled test data streams are used for continual model adaptation. Previous TTA schemes assume that the test samples are independent and identically distributed (i.i.d.), even though they are often temporally correlated (non-i.i.d.) in application scenarios, e.g., autonomous driving. We discover that most existing TTA methods fail dramatically under such scenarios. Motivated by this, we present a new test-time adaptation scheme that is robust against non-i.i.d. test data streams. Our novelty is mainly two-fold: (a) Instance-Aware Batch Normalization (IABN) that corrects normalization for out-of-distribution samples, and (b) Prediction-balanced Reservoir Sampling (PBRS) that simulates i.i.d. data stream from non-i.i.d. stream in a class-balanced manner. Our evaluation with various datasets, including real-world non-i.i.d. streams, demonstrates that the proposed robust TTA not only outperforms state-of-the-art TTA algorithms in the non-i.i.d. setting, but also achieves comparable performance to those algorithms under the i.i.d. assumption. Code is available at https://github.com/TaesikGong/NOTE.

## 1   Introduction

While deep neural networks (DNNs) have been successful in several applications, their performance degrades under distributional shifts between the training data and test data [32]. This distributional shift hinders DNNs from being widely deployed in many risk-sensitive applications, such as autonomous driving, medical imaging, and mobile health care, where new types of test data unseen during training could result in undesirable disasters. For instance, Tesla Autopilot has caused 12 "deaths" until recently [2]. To address this problem, *test-time adaptation* (TTA) aims to adapt DNNs to the target/unseen domain with only unlabeled test data streams, without any additional data acquisition or labeling cost. Recent studies reported that TTA is a promising, practical direction to mitigate distributional shifts [29, 33, 41, 4, 44].

Prior TTA studies typically assume (implicitly or explicitly) that a target test sample $\mathbf{x}_t$ at time $t$ and the corresponding ground-truth label $y_t$ (unknown to the learner) are independent and identically distributed (i.i.d.) following a target domain, i.e., $(\mathbf{x}_t, y_t)$ is drawn independently from a time-invariant distribution $P_{\mathcal{T}}(\mathbf{x}, y)$. However, the distribution of online test samples often changes across the time axis, i.e., $(\mathbf{x}_t, y_t) \sim P_{\mathcal{T}}(\mathbf{x}, y \mid t)$ in many applications; for instance, AI-powered self-driving car's object encounter will be dominated by cars while driving on the highway, but less dominated by them on downtown where other classes such as pedestrians and bikes are visible. In human activity recognition, some activities last for a short term (e.g., a fall down), whereas certain activities last longer (e.g., a sleep). Figure 1 illustrates that some data distributions in the real world, such as autonomous driving and human activity recognition, are often temporally correlated. Considering that most

36th Conference on Neural Information Processing Systems (NeurIPS 2022).

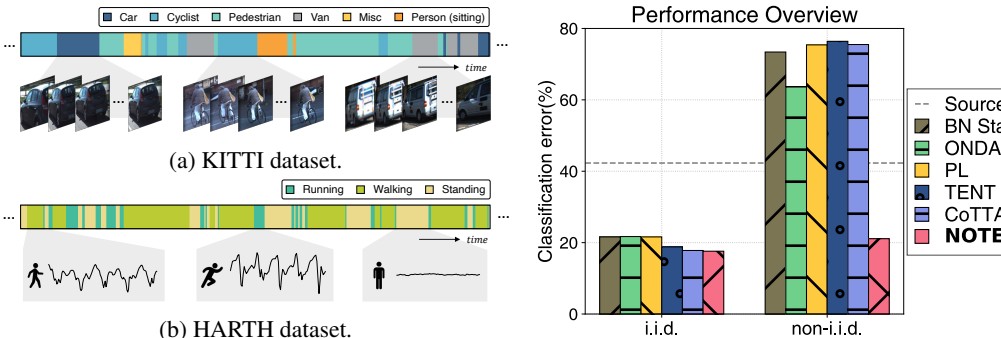

Figure 1: Illustration of test sample distributions along the time axis from real-world datasets: (a) autonomous driving (KITTI [9]) and (b) human activity recognition (HARTH [25]). They are temporally correlated.

Figure 2: Average classification error (%) of existing TTA methods and our method (NOTE) on CIFAR10-C [13]. The error rates significantly increase under the non-i.i.d. setting compared with the i.i.d. setting. Lower is better.

existing TTA algorithms simply use an incoming batch of test samples for adaptation [29, 33, 41, 44], the model might be biased towards these imbalanced samples under the temporally correlated test streams. Figure 2 compares the performance of the state-of-the-art TTA algorithms under the i.i.d. and non-i.i.d.[1] conditions. While the TTA methods perform well under the i.i.d. assumption, their errors increase under the non-i.i.d. case. Adapting to temporally correlated test data results in overfitting to temporal distributions, which in turn harms the generalization of the model.

Motivated by this, we present a NOn-i.i.d. TEst-time adaptation scheme, *NOTE*, that consists of two components: (a) Instance-Aware Batch Normalization (IABN) and (b) Prediction-Balanced Reservoir Sampling (PBRS). First, we propose a novel normalization layer, IABN, that eliminates the dependence on temporally correlated data for adaptation while being robust to distribution shifts. IABN detects out-of-distribution instances *sample by sample* and corrects via instance-aware normalization. The key idea of IABN is synthesizing Batch Normalization (BN) [16] with Instance Normalization (IN) [37] in a unique way; it calculates how different the learned knowledge (BN) is from the current observation (IN) and corrects the normalization by the deviation between IN and BN. Second, we present PBRS that resolves the problem of overfitting to non-i.i.d. samples by mimicking i.i.d. samples from non-i.i.d. streams. By utilizing predicted labels of the model, PBRS aims for both time-uniform sampling and class-uniform sampling from the non-i.i.d. streams and stores the 'simulated' i.i.d. samples in memory. With the i.i.d.-like batch in the memory, PBRS enables the model to adapt to the target domain without being biased to temporal distributions.

We evaluate NOTE with state-of-the-art TTA baselines [29, 33, 22, 27, 4, 44] on multiple datasets, including common TTA benchmarks (CIFAR10-C, CIFAR100-C, and ImageNet-C [13]) and real-world non-i.i.d. datasets (KITTI [9], HARTH [25], and ExtraSensory [38]). Our results suggest that NOTE not only significantly outperforms the baselines under non-i.i.d. test data, e.g., it achieves a 21.1% error rate on CIFAR10-C which is on average 15.1% lower than the state-of-the-art method [4], but also shows comparable performance even under the i.i.d. assumption, e.g., 17.6% error on CIFAR10-C where the best baseline [44] achieves 17.8% error. Our ablative study demonstrates the individual effectiveness of IABN and PBRS and further highlights their synergy when jointly used.

Finally, we summarize the key characteristics of NOTE. First, NOTE is a batch-free inference algorithm (requiring a single instance for inference), different from the state-of-the-art TTA algorithms [29, 33, 41, 4, 44] where a batch of test data is necessary for inference to estimate normalization statistics (mean and variance). Second, while some recent TTA methods leverage augmentations to improve performance at the cost of additional forwarding passes [35, 44], NOTE requires only a single forwarding pass. NOTE updates only the normalization statistics and affine parameters in IABN, which is, e.g., approximately 0.02% of the total trainable parameters in ResNet18 [12]. Third, NOTE's additional memory overhead is negligible. It merely stores predicted labels of test data to run PBRS. These characteristics make NOTE easy to apply to any existing AI system and particularly, are beneficial in latency-sensitive tasks such as autonomous driving and human health monitoring.

---

[1]We use the terms *temporally correlated* and *non-i.i.d.* interchangeably in the context of test-time adaptation.

## 2 Background

### 2.1 Problem setting: test-time adaptation with non-i.i.d. streams

**Test-time adaptation.** Let $\mathcal{D}_\mathcal{S} = \{\mathcal{X}^\mathcal{S}, \mathcal{Y}\}$ be the data from the source domain and $\mathcal{D}_\mathcal{T} = \{\mathcal{X}^\mathcal{T}, \mathcal{Y}\}$ be the data from the target domain to adapt to. Each data instance and the corresponding ground-truth label pair $(\mathbf{x}_i, y_i) \in \mathcal{X}^\mathcal{S} \times \mathcal{Y}$ in the source domain follows a probability distribution $P_\mathcal{S}(\mathbf{x}, y)$. Similarly, each target test sample and the corresponding label at test time $t$, $(\mathbf{x}_t, y_t) \in \mathcal{X}^\mathcal{T} \times \mathcal{Y}$, follows a probability distribution $P_\mathcal{T}(\mathbf{x}, y)$ where $y_t$ is unknown for the learner. The standard covariate shift assumption in domain adaptation is defined as $P_\mathcal{S}(\mathbf{x}) \neq P_\mathcal{T}(\mathbf{x})$ and $P_\mathcal{S}(y|\mathbf{x}) = P_\mathcal{T}(y|\mathbf{x})$ [32]. Unlike traditional domain adaptation that uses $\mathcal{D}_\mathcal{S}$ and $\mathcal{X}^\mathcal{T}$ collected beforehand for adaptation, test-time adaptation (TTA) continually adapts a pre-trained model $f_\theta(\cdot)$ from $\mathcal{D}_\mathcal{S}$, by utilizing only $\mathbf{x}_t$ obtained at test time $t$.

**TTA on non-i.i.d. streams.** Note that previous TTA mechanisms typically assume that each target sample $(\mathbf{x}_t, y_t) \in \mathcal{X}^\mathcal{T} \times \mathcal{Y}$ is independent and identically distributed (i.i.d.) following a time-invariant distribution $P_\mathcal{T}(\mathbf{x}, y)$. However, the data obtained at test time is non-i.i.d. in many scenarios. By non-i.i.d., we refer to distribution changes over time, i.e., $(\mathbf{x}_t, y_t) \sim P_\mathcal{T}(\mathbf{x}, y \,|\, t)$, which is a practical setting in many real world applications [46].

### 2.2 Batch normalization in TTA

Batch Normalization (BN) [16] is a widely-used training technique in deep neural networks as it reduces the internal covariant shift problem. Let $\mathbf{f} \in \mathbb{R}^{B \times C \times L}$ denote a batch of feature maps in general, where $B$, $C$, and $L$ denote the batch size, the number of channels, and the size of each feature map, respectively. Given the statistics of the feature maps for normalization, say mean $\boldsymbol{\mu}$ and variance $\boldsymbol{\sigma}^2$, BN is *channel-wise*, i.e., $\boldsymbol{\mu}, \boldsymbol{\sigma}^2 \in \mathbb{R}^C$ and computes:

$$\text{BN}(\mathbf{f}_{:,c,:}; \boldsymbol{\mu}_c, \boldsymbol{\sigma}_c^2) := \gamma \cdot \frac{\mathbf{f}_{:,c,:} - \boldsymbol{\mu}_c}{\sqrt{\boldsymbol{\sigma}_c^2 + \epsilon}} + \beta, \tag{1}$$

where $\gamma$ and $\beta$ are the affine parameters followed by the normalization, and $\epsilon > 0$ is a small constant for numerical stability.

Although a conventional way of computing BN in test-time is to set $\boldsymbol{\mu}$ and $\boldsymbol{\sigma}^2$ as those estimated from *training* (or source) data, say $\bar{\boldsymbol{\mu}}$ and $\bar{\boldsymbol{\sigma}}^2$, the state-of-the-art TTA methods based on adapting BN layers [29, 33, 41, 44] instead use the statistics computed directly from the recent test batch to de-bias distributional shifts at test-time, i.e.:

$$\hat{\boldsymbol{\mu}}_c := \frac{1}{BL} \sum_{b,l} \mathbf{f}_{b,c,l}, \text{ and } \hat{\boldsymbol{\sigma}}_c^2 := \frac{1}{BL} \sum_{b,l} (\mathbf{f}_{b,c,l} - \hat{\boldsymbol{\mu}}_c)^2. \tag{2}$$

This practice is simple yet effective under distributional shifts and is thus adopted in many recent TTA studies [29, 33, 41, 44]. Based on the test batch statistics, they often further adapt the affine parameters via entropy minimization of the model outputs [41] or update the entire parameters with self-training [44].

## 3 Method

In the same vein as previous work [29, 33, 41], we focus on adapting BN layers in the given model to perform TTA, and this includes essentially two approaches: (a) re-calibrating (or adapting) channel-wise statistics for normalization (instead of using those learned from training), and (b) adapting the affine parameters (namely, $\gamma$ and $\beta$) after the normalization with respect to a certain objective based on test samples, e.g., the entropy minimization of model outputs [41].

Under scenarios where test data are temporally correlated, however, naïvely adapting to the incoming batch of test samples [29, 33, 41, 44] could be problematic for both approaches: the batch is now more likely to (a) remove instance-wise variations that are actually useful to predict $y$, i.e., the "contents" rather than "styles" through normalization, and (b) include a bias in $p(y)$ rather than uniform, which can negatively affect the test-time adaptation objective such as entropy minimization.

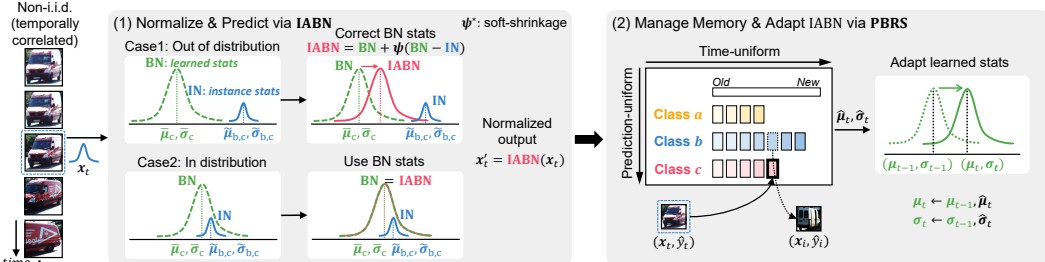

Figure 3: An overview of the proposed methodology: Instance-Aware Batch Normalization (IABN) and Prediction-Balanced Reservoir Sampling (PBRS). IABN aims to detect non-i.i.d. streams and in turn corrects the normalization for inference. PBRS manages data in a time- and prediction-uniform manner from non-i.i.d. data streams and gradually adapts IABNs with the balanced data afterward.

We propose two approaches to tackle each of the failure modes of adapting BN under temporal correlation. Our method consists of two components: (a) Instance-Aware Batch Normalization (IABN) (Section §3.1) to overcome the limitation of BN under distribution shift and (b) Prediction-Balanced Reservoir Sampling (PBRS) (Section §3.2) to combat with the temporal correlation of test batches. Figure 3 illustrates the overall workflow of NOTE with IABN and PBRS.

## 3.1 Instance-Aware Batch Normalization

As described in Section §2.2, recent TTA algorithms rely solely on the test batch to re-calculate BN statistics. We argue that this common practice does not successfully capture the feature statistics to normalize the feature map $\mathbf{f} \in \mathbb{R}^{B \times C \times L}$ under temporal correlation in the test batch $B$. In principle, standardizing a given feature map $\mathbf{f}_{:,c,:}$ by the statistics $\hat{\boldsymbol{\mu}}_c, \hat{\boldsymbol{\sigma}}_c^2$ computed across $B$ and $L$ is posited on premise that averaging information across $B$ can marginalize out uninformative instance-wise variations for predicting $y$. Under temporal correlation in $B$, however, this assumption is no longer valid, and averaging across $B$ may not fully de-correlate useful information in $\mathbf{f}_{:,c,:}$ from $\boldsymbol{\mu}_c$ and $\boldsymbol{\sigma}_c^2$.

In an attempt to bypass such an "over-whitening" effect of using $\hat{\boldsymbol{\mu}}_c$ and $\hat{\boldsymbol{\sigma}}_c^2$ in test-time under temporal correlation, we propose correcting normalization statistics on a *per-sample basis*: specifically, instead of completely switching from the original statistics of $(\bar{\boldsymbol{\mu}}, \bar{\boldsymbol{\sigma}}^2)$ into $(\hat{\boldsymbol{\mu}}_c, \hat{\boldsymbol{\sigma}}_c^2)$, our proposed *Instance-Aware Batch Normalization* (IABN) considers the *instance-wise* statistics $\tilde{\boldsymbol{\mu}}, \tilde{\boldsymbol{\sigma}}^2 \in \mathbb{R}^{B,C}$ of $\mathbf{f}$ similarly to Instance Normalization (IN) [37], namely:

$$\tilde{\boldsymbol{\mu}}_{b,c} := \frac{1}{L}\sum_l \mathbf{f}_{b,c,l} \text{ and } \tilde{\boldsymbol{\sigma}}_{b,c}^2 := \frac{1}{L}\sum_l (\mathbf{f}_{b,c,l} - \tilde{\boldsymbol{\mu}}_{b,c})^2. \tag{3}$$

We assume that $\tilde{\boldsymbol{\mu}}_{b,c}$ and $\tilde{\boldsymbol{\sigma}}_{b,c}^2$ follow the *sampling distribution* of a sample size $L$ in $\mathcal{N}(\bar{\boldsymbol{\mu}}, \bar{\boldsymbol{\sigma}}^2)$ as the population. Then the corresponding variances for the sample mean $\tilde{\boldsymbol{\mu}}_{b,c}$ and the sample variance $\tilde{\boldsymbol{\sigma}}_{b,c}^2$ can be calculated as:

$$s_{\tilde{\boldsymbol{\mu}},c}^2 := \frac{\bar{\boldsymbol{\sigma}}_c^2}{L} \text{ and } s_{\tilde{\boldsymbol{\sigma}}^2,c}^2 := \frac{2\bar{\boldsymbol{\sigma}}_c^4}{L-1}. \tag{4}$$

IABN corrects $(\bar{\boldsymbol{\mu}}, \bar{\boldsymbol{\sigma}}^2)$ only in cases when $\tilde{\boldsymbol{\mu}}_{b,c}$ (and $\tilde{\boldsymbol{\sigma}}_{b,c}^2$) significantly differ from $\bar{\boldsymbol{\mu}}_c$ (and $\bar{\boldsymbol{\sigma}}_c^2$). Specifically, we propose to use the following statistics for TTA:

$$\boldsymbol{\mu}_{b,c}^{\texttt{IABN}} := \bar{\boldsymbol{\mu}}_c + \psi(\tilde{\boldsymbol{\mu}}_{b,c} - \bar{\boldsymbol{\mu}}_c; \alpha s_{\tilde{\boldsymbol{\mu}},c}), \text{ and } (\boldsymbol{\sigma}_{b,c}^{\texttt{IABN}})^2 := \bar{\boldsymbol{\sigma}}_c^2 + \psi(\tilde{\boldsymbol{\sigma}}_{b,c}^2 - \bar{\boldsymbol{\sigma}}_c^2; \alpha s_{\tilde{\boldsymbol{\sigma}}^2,c}),$$

$$\text{where } \psi(x;\lambda) = \begin{cases} x - \lambda, & \text{if } x > \lambda \\ x + \lambda, & \text{if } x < -\lambda \\ 0, & \text{otherwise} \end{cases} \text{ is the soft-shrinkage function.} \tag{5}$$

$\alpha \geq 0$ is the hyperparameter of IABN that determines the confidence level of the BN statistics. A high value of $\alpha$ relies more on the learned statistics (BN), while a low value of $\alpha$ is in favor of the current statistics measured from the instance. Finally, the output of IABN can be described as:

**Algorithm 1** Prediction-Balanced Reservoir Sampling

---

**Input:** target stream $\mathbf{x}_t \sim P_{\mathcal{T}}(\mathbf{x}|t)$, memory bank $M$ of capacity $N$
1:  $M[i] \leftarrow \phi$ for $i = 1, \cdots N$; and $n[c] \leftarrow 0$ for $c \in \mathcal{Y}$
2:  **for** $t \in \{1, \cdots, T\}$ **do**
3:     $n[\hat{y}_t] \leftarrow n[\hat{y}_t] + 1$    // increase the number of samples encountered for the class
4:     $m[c] \leftarrow |\{(\mathbf{x}, y) \in M | y = c\}|$ for $c \in \mathcal{Y}$    // count instances per class in memory
5:     **if** $|M| < N$ **then**    // if memory is not full
6:         Add $(\mathbf{x}_t, \hat{y}_t)$ to $M$
7:     **else**
8:         $C^* \leftarrow \arg\max_{c \in \mathcal{Y}} m[c]$    // get majority class(es)
9:         **if** $\hat{y}_t \notin C^*$ **then**    // if the new sample is not majority         ▷ Prediction-Balanced
10:           Randomly pick $M[i] := (\mathbf{x}_i, \hat{y}_i)$ where $\hat{y}_i \in C^*$
11:           $M[i] \leftarrow (\mathbf{x}_t, \hat{y}_t)$    // replace it with a new sample
12:         **else**                                                     ▷ Reservoir Sampling
13:           Sample $p \sim \text{Uniform}(0, 1)$
14:           **if** $p < m[\hat{y}_t]/n[\hat{y}_t]$ **then**
15:             Randomly pick $M[i] := (\mathbf{x}_i, \hat{y}_i)$ where $\hat{y}_i = \hat{y}_t$
16:             $M[i] \leftarrow (\mathbf{x}_t, \hat{y}_t)$    // replace it with a new sample

---

$$\text{IABN}(\mathbf{f}_{b,c,:}; \bar{\boldsymbol{\mu}}_c, \bar{\boldsymbol{\sigma}}_c^2; \tilde{\boldsymbol{\mu}}_{b,c}, \tilde{\boldsymbol{\sigma}}_{b,c}^2) := \gamma \cdot \frac{\mathbf{f}_{b,c,:} - \boldsymbol{\mu}_{b,c}^{\texttt{IABN}}}{\sqrt{(\boldsymbol{\sigma}_{b,c}^{\texttt{IABN}})^2 + \epsilon}} + \beta. \tag{6}$$

Observe that IABN becomes IN and BN when $\alpha = 0$ and $\alpha = \infty$, respectively. If one chooses too small $\alpha \geq 0$, IABN may remove useful features, e.g., styles, of input (as with IN), which can degrade the overall classification (or regression) performance [30]. Hence, it is important to choose an appropriate $\alpha$. Nevertheless, we found that a good choice of $\alpha$ is not too sensitive across tested scenarios, where we chose $\alpha = 4$ for all experiments. This way, IABN can be robust to distributional shifts without the risk of eliminating crucial information to predict $y$.

### 3.2 Adaptation via Prediction-Balanced Reservoir Sampling

Temporally correlated distributions lead to an undesirable bias in $p(y)$, and thus adaptation with a batch of consecutive test samples negatively impacts the adaptation objective, such as entropy minimization [41]. To combat this imbalance, we propose Prediction-Balanced Reservoir Sampling (PBRS) that mimics i.i.d. samples from temporally correlated streams with the assistance of a small (e.g., a mini-batch size) memory. PBRS combines *time-uniform* sampling and *prediction-uniform* sampling to simulate i.i.d. samples from the non-i.i.d. streams. For time-uniform sampling, we adopt reservoir sampling (RS) [40], a proven random sampling algorithm to collect time-uniform data in a single pass on a stream without prior knowledge of the total length of data. For prediction-uniform sampling, we first use the predicted labels to compute the majority class(es) in the memory. We then replace a random instance of the majority class(es) with a new sample. We detail the algorithm of PBRS as a pseudo-code in Algorithm 1. We found that these two heuristics can effectively balance samples among both time and class axes, which mitigates the bias in temporally correlated data.

With the stored samples in the memory, we update the normalization statistics and affine parameters in the IABN layers. Note that IABN assumes $\tilde{\boldsymbol{\mu}}_{b,c}$ and $\tilde{\boldsymbol{\sigma}}_{b,c}^2$ follow the sampling distribution of $\mathcal{N}(\bar{\boldsymbol{\mu}}, \bar{\boldsymbol{\sigma}}^2)$ and corrects the normalization if $\tilde{\boldsymbol{\mu}}_{b,c}$ and $\tilde{\boldsymbol{\sigma}}_{b,c}^2$ are out of distribution. While IABN is resilient to distributional shifts to a certain extent, the assumption might not hold under severe distributional shifts. Therefore, we aim to find better estimates of $\bar{\boldsymbol{\mu}}, \bar{\boldsymbol{\sigma}}^2$ in IABN under distributional shifts via PBRS. Specifically, we update the normalization statistics, namely the means $\boldsymbol{\mu}$ and variances $\boldsymbol{\sigma}^2$, via exponential moving average: (a) $\boldsymbol{\mu}_t = (1 - m)\boldsymbol{\mu}_{t-1} + m \frac{N}{N-1} \hat{\boldsymbol{\mu}}_t$ and (b) $\boldsymbol{\sigma}_t^2 = (1 - m)\boldsymbol{\sigma}_{t-1}^2 + m \frac{N}{N-1} \hat{\boldsymbol{\sigma}}_t^2$ where $m$ is a momentum and $N$ is the size of the memory. We further optimize the affine parameters, scaling factor $\gamma$ and bias term $\beta$, via a single backward pass with entropy minimization, similar to a previous study [41]. These parameters account for only around 0.02% of the total trainable parameters in ResNet18 [12]. The IABN layers are adapted with the

$N$ samples in the memory every $N$ test samples. We set the memory size $N$ as 64 following the common batch size of existing TTA methods [33, 4, 41] to ensure a fair memory constraint.

## 3.3 Inference

NOTE infers each sample via a single forward pass with IABN layers. Note that NOTE requires only a single instance for inference, different from the state-of-the-art TTA methods [29, 33, 41, 4, 44] that require batches for every inference. Moreover, NOTE requires only one forwarding pass for inference, while multiple forward passes are required in other TTA methods that utilize augmentations [35, 44]. The batch-free single-forward inference of NOTE is beneficial in latency-sensitive tasks such as autonomous driving and human health monitoring. After inference, NOTE determines whether to store the sample and predicted label in the memory via PBRS.

## 4 Experiments

We implemented NOTE and the baselines via the PyTorch framework [31].[2] We ran all experiments with three random seeds and report the means and standard deviations. Additional experimental details, e.g., hyperparameters of the baselines and datasets, are specified in Appendix A.

**Baselines.** We consider the following baselines including the state-of-the-art test-time adaptation algorithms: **Source** evaluates the model trained from the source data directly on the target data without adaptation. Test-time normalization (**BN stats**) [29, 33] updates the BN statistics from a batch of test data. Online Domain Adaptation (**ONDA**) [27] adapts batch normalization statistics to target domains via a batch of target data with an exponential moving average. Pseudo-Label (**PL**) [22] optimizes the trainable parameters in BN layers via hard pseudo labels. We update the BN layers only in PL, as done in previous studies [41, 44]. Test entropy minimization (**TENT**) [41] updates the BN parameters via entropy minimization. Laplacian Adjusted Maximum-likelihood Estimation (**LAME**) [4] takes a more conservative approach; it modifies the classifier's output probability and not the internal parameters of the model itself. By doing so, it prevents the model parameters from over-adapting to the test batch. Continual test-time adaptation (**CoTTA**) [44] reduces the error accumulation by using weight-averaged and augmentation-averaged predictions. It avoids catastrophic forgetting by stochastically restoring a part of the neurons to the source pre-trained weights.

**Adaptation and hyperparameters.** We assume the model pre-trained with source data is available for TTA. In NOTE, we replaced BN with IABN during training. We set the test batch size as 64 and the adaptation epoch as one for adaptation, which is the most common setting among the baselines [33, 4, 41]. Similarly, we set the memory size $N$ as 64 and adapt the model every 64 samples in NOTE to ensure a fair memory constraint. We conduct online adaptation and evaluation, where the model is continually updated. For the baselines, we adopt the best values for the hyperparameters reported in their papers or the official codes. We followed the guideline to tune the hyperparameters when such a guideline was available [44]. We use fixed values for the hyperparameters of NOTE, soft-shrinkage width $\alpha = 4$ and exponential moving average momentum $m = 0.01$, and update the affine parameters via the Adam optimizer [18] with a learning rate of $l = 0.0001$ unless specified. We detailed hyperparameter information of the baselines in Appendix A.1.

**Datasets.** We use CIFAR10-C, CIFAR100-C, and ImageNet-C [13] datasets that are common TTA benchmarks for evaluating the robustness to corruptions [29, 33, 41, 44, 4]. Both CIFAR10/CIFAR100 [19] have 50,000/10,000 training/test data. ImageNet [7] has 1,281,167/50,000 training/test data. CIFAR10/CIFAR100/ImageNet have 10/100/1,000 classes, respectively. CIFAR10-C/CIFAR100-C/ImageNet-C apply 15 types of corruption to CIFAR10/CIFAR100/ImageNet test data. Similar to previous studies [29, 33, 41, 44], we use the most severe corruption level of 5. We use ResNet18 [12] as the backbone network and pre-trained it on the clean training data. Following prior studies [23, 15, 43, 42], we adopt Dirichlet distribution to generate synthetic non-i.i.d. test streams from the originally i.i.d. CIFAR10/100 data. The details are provided in Appendix A.2. We vary the Dirichlet concentration parameter $\delta$ to simulate diverse streams and visualize the resulting data in Figure 4. We use $\delta = 0.1$ as the default value unless specified. For ImageNet, we sort the test stream

---

[2] https://github.com/TaesikGong/NOTE

Table 1: Average classification error (%) and their corresponding standard deviations on CIFAR10-C/100-C and ImageNet-C under temporally correlated (non-i.i.d.) and uniformly distributed (i.i.d.) test data stream. **Bold** fonts indicate the lowest classification errors, while Red fonts show performance degradation after adaptation. Values encompassed by parentheses refer to NOTE used directly with test batches (without using PBRS). Averaged over three runs.

| | Temporally correlated test stream | | | | Uniformly distributed test stream | | | |
|---|---|---|---|---|---|---|---|---|
| Method | CIFAR10-C | CIFAR100-C | ImageNet-C | Avg | CIFAR10-C | CIFAR100-C | ImageNet-C | Avg |
| Source | 42.3 ± 1.1 | 66.6 ± 0.1 | 86.1 ± 0.0 | 65.0 | 42.3 ± 1.1 | 66.6 ± 0.1 | 86.1 ± 0.0 | 65.0 |
| BN Stats [29] | 73.4 ± 1.3 | 65.0 ± 0.3 | 96.9 ± 0.0 | 78.5 | 21.6 ± 0.4 | 46.6 ± 0.2 | 76.0 ± 0.0 | 48.1 |
| ONDA [27] | 63.6 ± 1.0 | 49.6 ± 0.3 | 89.0 ± 0.0 | 67.4 | 21.7 ± 0.4 | 46.5 ± 0.1 | 75.9 ± 0.0 | 48.0 |
| PL [22] | 75.4 ± 1.8 | 66.4 ± 0.4 | 98.9 ± 0.0 | 80.2 | 21.6 ± 0.2 | 43.1 ± 0.3 | 74.4 ± 0.2 | 46.4 |
| TENT [41] | 76.4 ± 2.7 | 66.9 ± 0.6 | 96.9 ± 0.0 | 80.1 | 18.8 ± 0.2 | **40.3 ± 0.2** | 76.0 ± 0.0 | 45.0 |
| LAME [4] | 36.2 ± 1.3 | 63.3 ± 0.3 | 82.7 ± 0.0 | 60.7 | 44.1 ± 0.5 | 68.8 ± 0.1 | 86.3 ± 0.0 | 66.4 |
| CoTTA [44] | 75.5 ± 0.7 | 64.2 ± 0.2 | 97.0 ± 0.0 | 78.9 | 17.8 ± 0.3 | 44.3 ± 0.2 | 71.5 ± 0.0 | 44.6 |
| NOTE | **21.1 ± 0.6** | **47.0 ± 0.1** | **80.6 ± 0.1** | **49.6** | 20.1 ± 0.5 (**17.6 ± 0.3**) | 46.4 ± 0.0 (41.0 ± 0.2) | **70.3 ± 0.0** (71.7 ± 0.0) | 45.6 (**43.4**) |

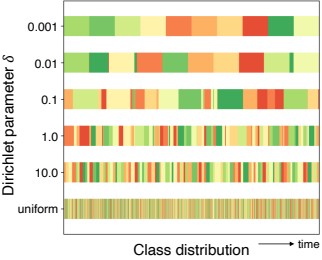

Figure 4: Illustration of synthetic non-i.i.d. streams sampled from Dirichlet distribution varying $\delta$ on CIFAR10-C. *uniform* denotes an i.i.d. condition. The lower the $\delta$, the more temporally correlated the distribution.

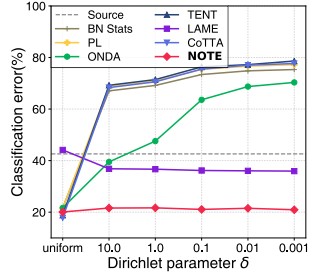

(a) Effect of Dirichlet concentration parameter $\delta$.

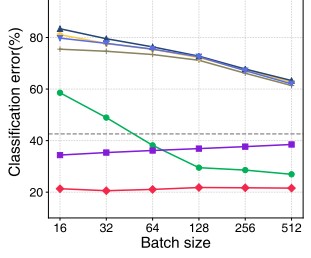

(b) Effect of batch size.

Figure 5: Average classification error (%) under the non-i.i.d. setting with CIFAR10-C dataset. We vary (a) the Dirichlet concentration parameter $\delta$ to investigate the impact of the degree of temporal correlation and (b) batch size to understand the behaviors of the TTA methods. Averaged over three runs. Lower is better.

as the number of test samples per class is not enough for generating temporally correlated streams via Dirichlet distribution. We additionally provide an experiment with MNIST-C data [28] in the appendix, which shows similar takeaways to our experiments with CIFAR10-C/CIFAR100-C/ImageNet-C.

**Overall result.** Tables 1 shows the result under the temporally correlated (non-i.i.d.) data and the uniform (i.i.d.) data, respectively. We observe significant performance degradation in the baselines under the temporally correlated setting. For BN Stats, PL, TENT, and CoTTA, this degradation is particularly due to the dependence on the test batch for the re-calculation of the BN statistics. Updating the batch statistics from test data via exponential moving average (ONDA) also suffers from the temporally correlated data. This indicates relying on the test batch for re-calculating the BN statistics indeed cancels out meaningful instance-wise variations under temporal correlation. Interestingly, LAME works better in the non-i.i.d. setting than in the i.i.d. setting, which is consistent with previous reports [4]. The primary reason is, as stated by the authors, it "discourages deviations from the predictions of the pre-trained model," and thus it "does not noticeably help in i.i.d and class-balanced scenarios."

NOTE achieves on average 11.1% improvement over the best baseline (LAME) under the non-i.i.d. setting. With the i.i.d. assumption, NOTE still achieves comparable performance to the baselines. When we know the target samples are i.i.d., we can simply use the test batch without using PBRS. For this variant version of NOTE, we update IABN with incoming test batches directly using a ten times higher learning rate of 0.001 following previous work [41, 44]. We report the result of the variant

Table 2: Average classification error (%) and their corresponding standard deviations on three real test data streams: KITTI, HARTH, and ExtraSensory. **Bold** fonts indicate the lowest classification errors, while Red fonts show performance degradation after adaptation. Averaged over three runs.

| | Real test stream | | | |
|---|---|---|---|---|
| Method | KITTI | HARTH | ExtraSensory | Avg |
| Source | 12.3 ± 2.3 | 62.6 ± 8.5 | 50.2 ± 2.2 | 41.7 |
| BN Stats [29] | 35.4 ± 0.5 | 68.6 ± 1.1 | 56.0 ± 0.9 | 53.4 |
| ONDA [27] | 26.3 ± 0.5 | 69.3 ± 1.1 | 48.2 ± 1.5 | 47.9 |
| PL [22] | 39.0 ± 0.3 | 64.8 ± 0.6 | 56.0 ± 0.9 | 53.3 |
| TENT [41] | 39.6 ± 0.2 | 64.1 ± 0.7 | 56.0 ± 0.8 | 53.2 |
| LAME [4] | 11.3 ± 2.9 | 61.0 ± 10.0 | 50.7 ± 2.7 | 41.0 |
| CoTTA [44] | 35.4 ± 0.6 | 68.7 ± 1.1 | 56.0 ± 0.9 | 53.4 |
| NOTE | **10.9 ± 3.6** | **51.0 ± 5.6** | **45.4 ± 2.6** | **35.8** |

version of NOTE in the parentheses, which achieves on average 2.2% improvement further when the i.i.d. assumption is known.

**Effect of the degree of temporal correlation.** We also investigate the effect of the degree of temporal correlation for TTA algorithms. Figure 5a shows the result. The lower $\delta$ is, the severer the temporal correlation becomes. The error rates of most of the baselines deteriorate as $\delta$ decreases, which shows that the existing TTA baselines are susceptible to temporally correlated data. NOTE shows consistent performance among all $\delta$ values, indicating its robustness under temporal correlation.

**Effect of batch size.** While we experiment with a widely-used value of 64 as the batch size (or memory size in NOTE), one might be curious about the impact of batch size under temporally correlated streams. Figure 5b shows the result with six different batch sizes. As shown, NOTE is not much affected by the batch size, while most of the baselines recover performance degradation as the batch size increases. This is because a higher batch size has a better chance of adaptation with balanced samples under temporally correlated streams. Increasing the batch size, however, mitigates temporal correlation at the expense of inference latency and adaptation speed.

## 4.1 Real-distributions with domain shift

**Datasets.** We evaluate NOTE under three real-world distribution datasets: object detection in autonomous driving (KITTI [9]), human activity recognition (HARTH [25]), and user behavioral context recognition (ExtraSensory [38]). Additional dataset-specific details are in Appendix A.2.

KITTI is a well-known autonomous driving dataset that provides consecutive frames that contains natural temporal correlation in driving contexts. We adapted from KITTI to KITTI-Rain [11] - a dataset that converted KITTI images to rainy images. This contains 7,481/7,800 train/test samples with nine classes. We use ResNet50 [12] pre-trained on ImageNet [8] as the backbone network.

HARTH was collected from 22 users in free-living environments for seven days. Each user was equipped with two three-axial Axivity AX3 accelerometers for recording human activities. We use 15 users collectively as the source domain and the remaining seven users as each target domain, which entails natural domain shifts from source users to target users as different physical conditions make domain shifts across users. HARTH contains 82,544/39,377 train/test samples with 12 classes. We report the average error over all target domains. We use four one-dimensional convolutional layers followed by one fully-connected layer as the backbone network for HARTH.

The Extrasensory dataset collected users' own smartphone sensory data (motion sensors, audio, etc.) in the wild for seven days, aiming to capture people's authentic behaviors in their regular activities. We use 16 users as the source domain and seven users as target domains. ExtraSensory includes 17,777/4,862 train/test data with five classes. For ExtraSensory, we use two one-dimensional convolutional layers followed by one fully-connected layer as the backbone network. For both HARTH and ExtraSensory models, a single BN layer follows each convolutional layer.

Table 3: Average classification error (%) and corresponding standard deviations of varying ablation settings on CIFAR10-C/100-C under temporally correlated (non-i.i.d.) and uniformly distributed (i.i.d.) test data stream. **Bold** fonts indicate the lowest classification errors. Averaged over three runs.

| Method | Temporally correlated test stream | | | Uniformly distributed test stream | | |
| --- | --- | --- | --- | --- | --- | --- |
| | CIFAR10-C | CIFAR100-C | Avg | CIFAR10-C | CIFAR100-C | Avg |
| Source | 42.3 ± 1.1 | 66.6 ± 0.1 | 54.4 | 42.3 ± 1.1 | 66.6 ± 0.1 | 54.4 |
| IABN | 24.6 ± 0.6 | 54.5 ± 0.1 | 39.5 | 24.6 ± 0.6 | 54.5 ± 0.1 | 39.5 |
| PBRS | 27.5 ± 1.0 | 51.7 ± 0.2 | 39.6 | 25.8 ± 0.2 | 51.3 ± 0.1 | 38.5 |
| IABN+RS | **20.5 ± 1.5** | 48.2 ± 0.2 | 34.3 | 20.7 ± 0.6 | 48.3 ± 0.3 | 34.5 |
| IABN+PBRS | 21.1 ± 0.6 | **47.0 ± 0.1** | **34.0** | **20.1 ± 0.5** | **46.4 ± 0.0** | **33.2** |

**Result.** Table 2 shows the result for the real-world datasets. The overall trend is similar to the temporal correlation experiments with CIFAR10-C/CIFAR100-C/ImageNet-C datasets, which indicates that the real-world datasets are indeed temporally correlated. NOTE consistently reduces errors after adaptation under real-world distributions. We believe this demonstrates NOTE is a promising method to be utilized in various real-world ML applications with distributional shifts. We illustrate real-time classification error changes for real-world datasets in the appendix.

## 4.2 Ablation study

We conduct an ablative study to further investigate the individual components' effectiveness. Table 3 shows the result under both i.i.d. and non-i.i.d. settings. Using IABN alone significantly reduces error rates over Source, demonstrating the effectiveness of correcting normalization for out-of-distribution samples. Using PBRS with BN shows comparable improvement with the IABN-only result. Note that there is only a marginal gap (around 1%) between the non-i.i.d. and i.i.d. results in PBRS. This indicates that PBRS could effectively simulate i.i.d. samples from non-i.i.d. streams. The joint use of IABN and PBRS outperforms using either of them, meaning that PBRS provides IABN with better estimates for the normalizing operation. In addition, PBRS is better than Reservoir Sampling (RS) that has been a strong baseline in continual learning [17, 5]. This shows storing prediction-balanced sampling in addition to time-uniform sampling leads to better adaptation in TTA. We also investigated the joint use of IN and PBRS with the combination of IABN and PBRS on CIFAR100-C, and the result shows that IABN+PBRS (47.0%) achieves a lower error rate than IN+PBRS (52.5%) on CIFAR100-C under temporal correlation.

## 5 Related work

**Test-time adaptation.** Test-time adaptation (TTA) attempts to overcome distributional shifts with test data without the cost of data acquisition or labeling. TTA adapts to the target domain with only test data on the fly. Most existing TTA algorithms rely on a batch of test samples to adapt [29, 33, 41, 44] to re-calibrate BN layers on the test data. Simply using the statistics of a test batch in BN layers improves the robustness under distributional shifts [29, 33]. ONDA [27] updates the BN statistics with test data via exponential moving average. TENT [41] further updates the scaling and bias parameters in BN layers via entropy minimization.

Latest TTA studies consider distribution changes of test data [4, 44]. LAME [4] corrects the output probabilities of a classifier rather than tweaking the model's inner parameters. By restraining the model from over-adapting to the test batch, LAME allows the model to be more robust under non-i.i.d. scenarios. However, LAME does not have noticeable performance gains in class-balanced, standard i.i.d. scenarios. The primary reason is, as stated by the authors, it "discourages deviations from the predictions of the pre-trained model," and thus it "does not noticeably help in i.i.d and class-balanced scenarios." CoTTA [44] aims to adapt to continually changing target environments via a weight-averaged teacher model, weight-averaged augmentations, and stochastic restoring. However, CoTTA assumes i.i.d. test data within each domain and updates the entire model which increases computational costs.

There also exist works [35, 24] utilizing domain-specific self-supervision to resolve the distribution shift with test data, but are complementary to ours, i.e., we can also optimize the self-supervised loss

instead of the entropy loss, and not applicable to our setups of real test data streams as designing good self-supervision for these domains is highly non-trivial.

**Replay memory.** Replay memory is one of the major approaches in continual learning; it manages a buffer to replay previous data for future learning to prevent catastrophic forgetting. Reservoir sampling [40] is a random sampling algorithm that collects time-uniform samples from unknown sample streams with a single pass, and it has been proven to be a strong baseline in continual learning [17, 5]. GSS [1] stores samples to a memory in a way that maximizes the gradient direction among those samples. A recent study modifies reservoir sampling to balance classes under imbalanced data when the labels are given [6]. Our memory management scheme (PBRS) is inspired by these studies to prevent catastrophic forgetting in test-time adaptations.

## 6 Discussion and conclusion

This paper highlights that real-world distributions often change across the time axis, and existing test-time adaptation algorithms mostly suffer from the non-i.i.d. test data streams. To address this problem, we present a NOn-i.i.d. TEst-time adaptation algorithm, NOTE. Our experiments evaluated robustness under corruptions and domain adaptation on real-world distributions. The results demonstrate that NOTE not only outperforms the baselines under the non-i.i.d./real distribution settings, but it also shows comparable performance under the i.i.d. assumption. We believe that the insights and findings from this study are a meaningful step toward the practical impact of the test-time adaptation paradigm.

**Limitations.** NOTE and most state-of-the-art TTA algorithms [29, 22, 27, 33, 41, 44] assume that the backbone networks are equipped with BN (or IABN) layers. While BN is a widely-used component in deep learning, several architectures, such as LSTMs [14] and Transformers [39], do not embed BN layers. A recent study uncovered that BN is advantageous in Vision Transformers [45], showing potential room to apply our idea to architectures without BN layers. However, more in-depth studies are necessary to identify the actual applicability of BN (or IABN) to those architectures. While LAME [4] is applicable to models without BN, its limitation is the performance drop in i.i.d. scenarios, as shown in both its paper and our evaluation. While NOTE shows its effectiveness in both non-i.i.d and i.i.d. scenarios, a remaining challenge is to design an algorithm that generalizes to any architecture. We believe the findings and contributions of our work could give valuable insights to future endeavors on this end.

**Potential negative societal impacts.** As TTA relies on unlabeled test samples and changes the model accordingly, the model is exposed to potential data-driven biases after adaptation, such as fairness issues [3] and adversarial attacks [36]. In some sense, the utility of TTA comes at the expense of exposure to threats. This vulnerability is another crucial problem that both ML researchers and practitioners need to take into consideration. In addition, TTA entails additional computations for adaptation with test data, which may have negative impacts on environments, e.g., increasing electricity consumption and carbon emissions [34]. Nevertheless, we believe NOTE would not exacerbate this issue as it is computationally efficient as mentioned in Section §1.

## Acknowledgments and Disclosure of Funding

We thank the anonymous reviewers for their constructive feedback and suggestions to improve this paper. This work was supported in part by the National Research Foundation of Korea (NRF) grant funded by the Korea government (MSIP) (No.NRF-2020R1A2C1004062) and Center for Applied Research in Artificial Intelligence (CARAI) grant funded by Defense Acquisition Program Administration (DAPA) and Agency for Defense Development (ADD) (UD190031RD).

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
