# Appendix

## NOTE: Robust Continual Test-time Adaptation Against Temporal Correlation

## A  Experimental details

For all the experiments in the paper, we used three different random seeds (0, 1, 2) and reported the average errors (and standard deviations). We ran our experiments on NVIDIA GeForce RTX 3090 GPUs.

### A.1  Baseline details

We referred to the official implementations of the baselines. We use the reported best hyperparameters from their paper or code. We further tuned hyperparameters if there exists a hyperparameter selection guideline. Here, we provide additional details of the baseline implementations, including hyperparameters.

**PL.**  Following the previous studies [41, 44], we update the BN layers only in PL. We set the learning rate as $LR = 0.001$ as the same as [41].

**ONDA.**  ONDA [27] has two hyperparameters, the update frequency $N$ and the decay of the moving average $m$. The authors set $N = 10$ and $m = 0.1$ as the default values throughout the experiments, and we follow this choice unless specified.

**TENT.**  TENT [41] set the learning rate as $LR = 0.001$ for all datasets except for ImageNet, and we follow this choice. We referred to the official code[1] for implementing TENT.

**LAME.**  LAME [4] needs an affinity matrix and has hyperparameters related to it. We follow the authors' hyperparameter selection specified in the paper and their official code. Namely, we use the kNN affinity matrix with the value of k set as 5. We referred to the official code[2] for implementing LAME.

**CoTTA.**  CoTTA [44] has three hyperparameters, augmentation confidence threshold $p_{th}$, restoration factor $p$, and exponential moving average (EMA) factor $m$. We follow the authors' choice for restoration factor ($p = 0.01$) and EMA factor ($\alpha = 0.999$). For the augmentation confidence threshold, the authors provide a guideline to choose it, using 5% quantile for the softmax predictions' confidence on the source domains. We follow this guideline, which results in $p_{th} = 0.92$ for MNIST-C and CIFAR10-C, $p_{th} = 0.72$ for CIFAR100-C, and $p_{th} = 0.55$ for KITTI. For 1D time-series datasets (HARTH and ExtraSensory), the authors do not provide augmentations, and it is non-trivial to select appropriate augmentations for them. We thus do not use augmentations for these datasets. We referred to the official code[3] for implementing CoTTA.

### A.2  Dataset details

#### A.2.1  Robustness to corruptions

**MNIST-C.**  MNIST-C [28] applies 15 corruptions to the MNIST [21] dataset. Specifically, the corruptions include Shot Noise, Impulse Noise, Glass Blur, Motion Blur, Shear, Scale, Rotate, Brightness, Translate, Stripe, Fog, Spatter, Dotted Line, Zigzag, and Canny Edges, as illustrated in Figure 1. Note that the result of this dataset is included only in the supplementary material. In total, MNIST-C has 60,000 clean training data and 150,000 corrupted test data (10,000 for each corruption type). We use ResNet18 [12] as the backbone network. We train it on the clean training

---

[1] https://github.com/DequanWang/tent
[2] https://github.com/fiveai/LAME
[3] https://github.com/qinenergy/cotta

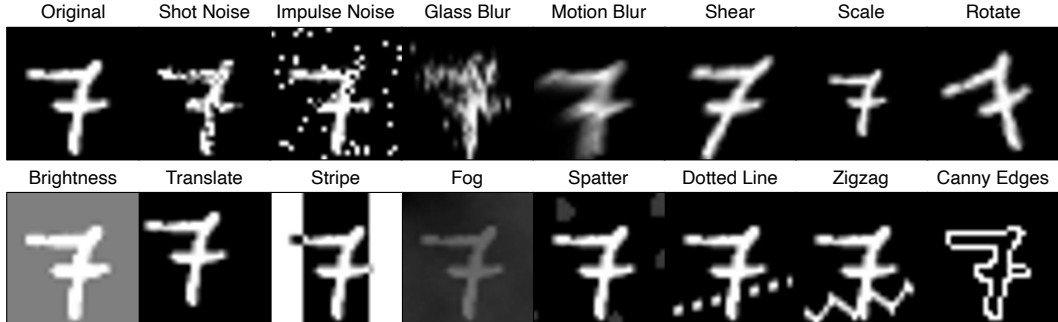

Figure 1: Illustration of the 15 corruption types in the MNIST-C dataset.

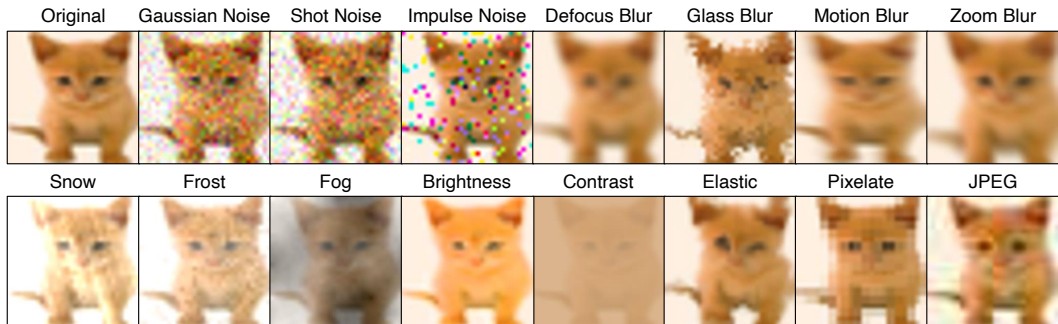

Figure 2: Illustration of the 15 corruption types in the CIFAR10-C/CIFAR100-C/ImageNet-C dataset.

data to generate source models, using stochastic gradient descent with momentum=0.9 and cosine annealing learning rate scheduling [26] for 100 epochs with an initial learning rate of 0.1.

**CIFAR10-C/CIFAR100-C.** CIFAR10-C/CIFAR100-C [13] are common TTA benchmarks for evaluating the robustness to corruptions [29, 33, 41, 44]. Both CIFAR10/CIFAR100 [19] have 50,000/10,000 training/test data. CIFAR10/CIFAR100 have 10/100 classes, respectively. CIFAR10-C/CIFAR100-C apply 15 types of corruptions to CIFAR10/CIFAR100 test data: Gaussian Noise, Shot Noise, Impulse Noise, Defocus Blur, Frosted Glass Blur, Motion Blur, Zoom Blur, Snow, Frost, Fog, Brightness, Contrast, Elastic Transformation, Pixelate, and JPEG Compression, as illustrated in Figure 2. We use the most severe corruption level of 5, similar to previous studies [29, 33, 41, 44]. This results in a total of 150,000 test data for CIFAR10-C/CIFAR100-C, respectively. We use ResNet18 [12] as the backbone network. We train it on the clean training data to generate source models, using stochastic gradient descent with momentum=0.9 and cosine annealing learning rate scheduling [26] for 200 epochs with an initial learning rate of 0.1 and a batch size of 128.

**ImageNet-C.** ImageNet-C is another common TTA benchmark for evaluating the robustness to corruptions [29, 33, 41, 44, 4]. ImageNet [7] has 1,281,167/50,000 training/test data. ImageNet-C applies the same 15 types of corruption used in CIFAR10-C and CIFAR100-C. We use a pre-trained ResNet18 [12] on ImageNet training data and fine-tune it by replacing BN layers with IABN layers on the clean ImageNet training data. For fine-tuning, we use stochastic gradient descent with momentum=0.9 for 30 epochs with a fixed learning rate of 0.001 and a batch size of 256.

**Temporally correlated streams via Dirichlet distribution.** Note that most public vision datasets are not time-series data, and existing TTA studies usually shuffled the order of these datasets resulting in i.i.d. streams, which might be unrealistic in real-world scenarios. To simulate non-i.i.d. streams from these "static" datasets, we utilize Dirichlet distribution that is widely used to simulate non-i.i.d. settings. [23, 15, 43, 42] Specifically, we simulate a non-i.i.d partition for $T$ tokens on $C$ classes. For each class $c$, we draw a $T$-dimensional vector $\mathbf{q}_c \sim Dir(\delta \mathbf{p})$, where $Dir(\cdot)$ denotes the Dirichlet distribution, $\mathbf{p}$ is a prior class distribution over $T$ classes, and $\delta > 0$ is a concentration parameter. We assign data from each class to each token $t$, following proportion $\mathbf{q}_c[n]$. To simulate the nature of

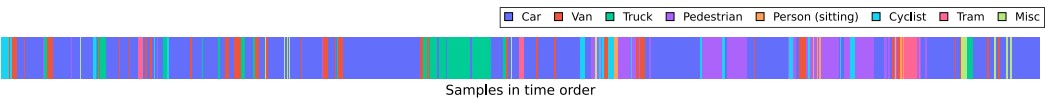

Car ■ Van ■ Truck ■ Pedestrian ■ Person (sitting) ■ Cyclist ■ Tram ■ Misc

Samples in time order

(a) Visualization of the class distribution in the entire KITTI dataset.

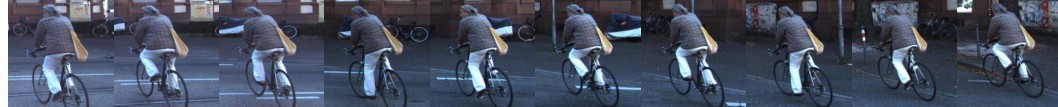

(b) Original data with an interval of three frames.

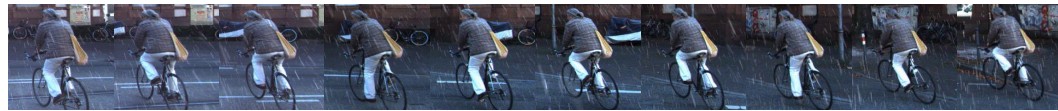

(c) Rain data with an interval of three frames.

Figure 3: Illustration of the test stream of the KITTI dataset. We apply a 200mm/hr rain intensity to the original data.

real-world online data where sequences are temporally correlated, and data from the same classes appear multiple times (e.g., walking, jogging, and then walking, see Figure 4 and 5 for illustrations), we concatenate the generated $T$ tokens to create a synthetic non-i.i.d. sequential data. We use $\delta = 0.1$ as the default value if not specified.

### A.2.2 Real-distributions with domain shift

The following illustrates the summary and preprocessing steps of datasets collected in the real world or have a resemblance to class distributions in the real world.

**KITTI, KITTI-Rain**. KITTI [9] is a well-known dataset used in numerous tasks such as object detection, object tracking, depth estimation, etc. It must be emphasized that the dataset was collected by driving around the city, in rural areas and on highways, which captures the real-world distribution. From the available tasks, we select the object tracking task; to utilize its temporal correlation. In order to reduce the task to a single image classification task, we crop each frame with respect to the largest bounding box. Domain gap is introduced through synthetic generation of corresponding "rainy" frames, hereby denoted as KITTI-Rain [11]. KITTI-Rain is generated via a two-step procedure: (1) generation of a depth-map estimation of each frame, and (2) generation of rainy images from the vanilla frame and its corresponding depth map, as described in [11]. For the depth map generation, we used Monodepth [10], and for rainy image generation, we used the source code available in [11]. The rain intensity is set to 200mm/hr for training and testing. The final source domain consists of 7,481 samples, and each of the target domains consists of 7,800 samples. We use ResNet50 [12] pre-trained on ImageNet [8] as the backbone network. We fine-tune it on the KITTI training data to generate source models, using the Adam optimizer [18] and cosine annealing learning rate scheduling [26] for 50 epochs with an initial learning rate of $0.1$ and a batch size of 64.

**HARTH**. Human Activity Recognition Trondheim dataset [25] was collected from 22 users, with two three-axial Axivity AX3 accelerometers, each attached to the subject's thigh and lower back. HARTH was also collected in a free-living environment and labeled through recorded video. We set the source domain as the accelerometer data collected from the back (15 users), and set the target domain as one collected from the thigh (from the remaining seven users). We deem such a setting to be natural, for one of the most dominant forms of domain shift in wearable sensory data is by the positioning of sensors on the human body [20]. We use a window size of 50 and min-max scaled (0-1) the data, following the original paper [25]. The final source domain consists of 82,544 samples, and each of the seven target domains consists of {S008: 8,140, S018: 6,241, S019: 5,846, S021: 5,910, S022: 6,448, S028: 3,271, S029: 3,521} samples. We use four one-dimensional convolutional layers followed by one fully-connected layer as the backbone network. We train it on the source data to generate source models, using stochastic gradient descent with momentum=0.9 for 100 epochs and

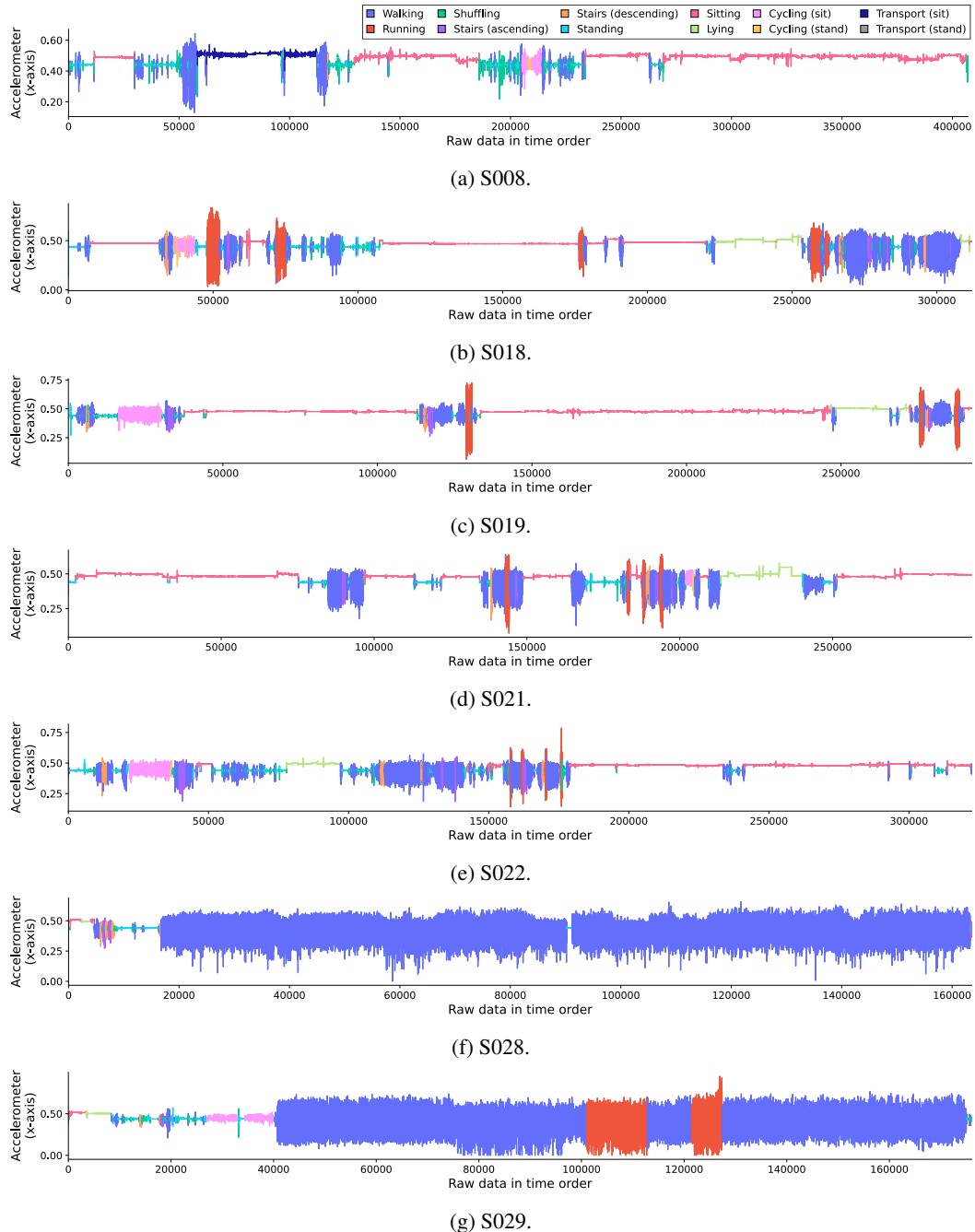

Figure 4: Illustration of the target streams of the HARTH dataset. We specify x-axis accelerometer values only.

cosine annealing learning rate scheduling [26] with an initial learning rate of 0.1 and a batch size of 64.

**ExtraSensory**. Extrasensory dataset [38] was collected from 60 users with the user's own smartphones over a seven-day period in the wild, i.e., data was collected from users who engaged in their regular natural behavior. As there were no constraints on the subject's activity, the distribution varied from user to user. We select the five most frequently occurred, mutually exclusive activities (lying down, sitting, walking, standing, running) and omit other labels. We further process the data to only those consisting of the following sensor modalities - accelerometer, gyroscope, magnetometer, and audio. We used a window size of five, with no overlap, and standardly scaled

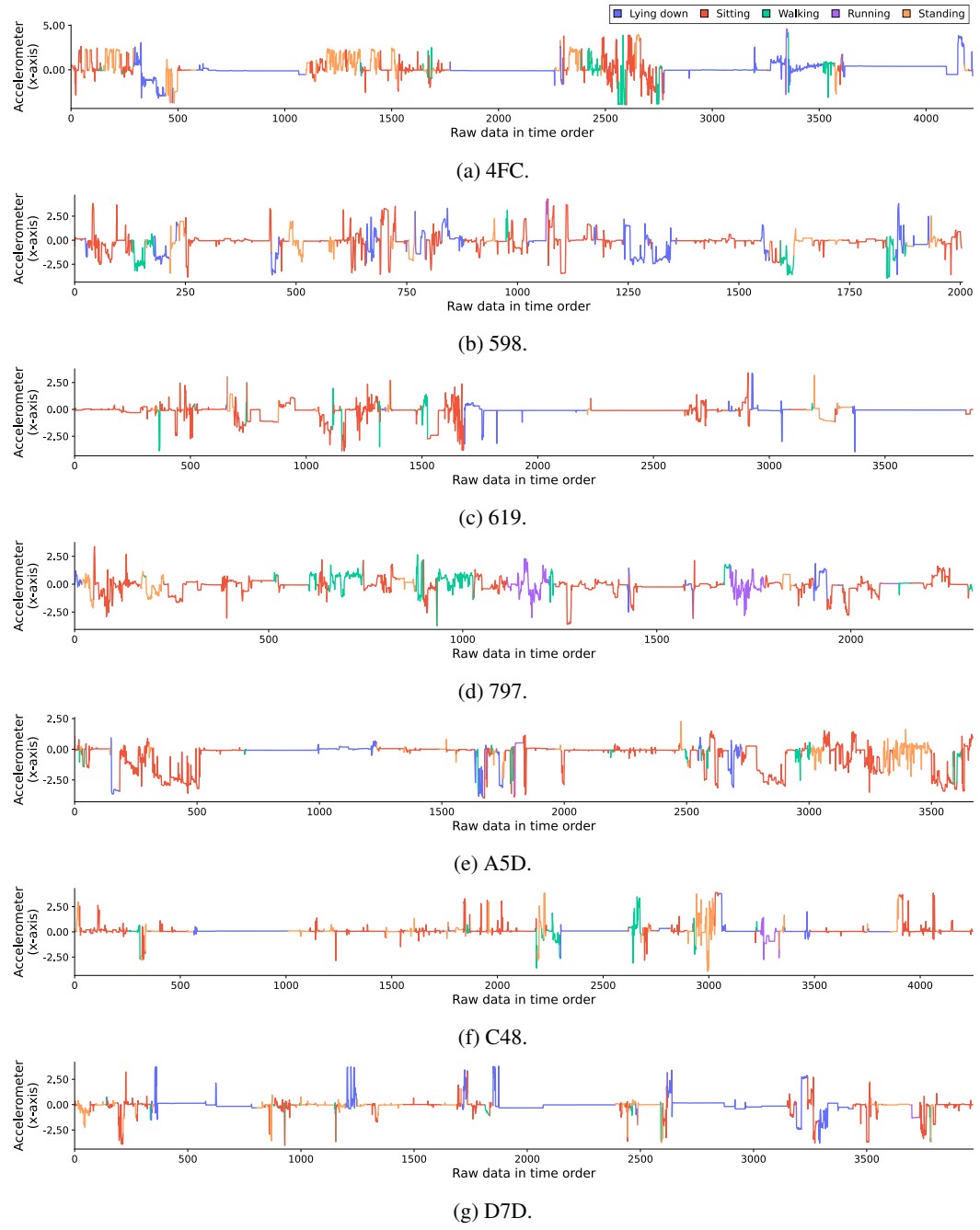

Figure 5: Illustration of the target streams of the Extrasensory dataset. We specify x-axis accelerometer values only. Due to the length of the name of each domain, denoted here with the first three characters.

the datasets. After the pre-processing step, 23 users were left, 16 of them were used as source domains, and seven of them were used as target domains. The final source domain consists of 17,777 samples, and each of the seven target domains consists of {4FC32141-E888-4BFF-8804-12559A491D8C: 844, 59818CD2-24D7-4D32-B133-24C2FE3801E5: 401, 61976C24-1C50-4355-9C49-AAE44A7D09F6: 776, 797D145F-3858-4A7F-A7C2-A4EB721E133C: 463, A5CDF89D-02A2-4EC1-89F8-F534FDABDD96 : 734, C48CE857-A0DD-4DDB-BEA5-3A25449B2153 : 850, D7D20E2E-FC78-405D-B346-DBD3FD8FC92B: 794} samples. We use two one-dimensional convolutional layers followed by one fully-connected layer as the backbone network. We train it on the source data to generate source models, using stochastic gradient descent with momentum=0.9 for

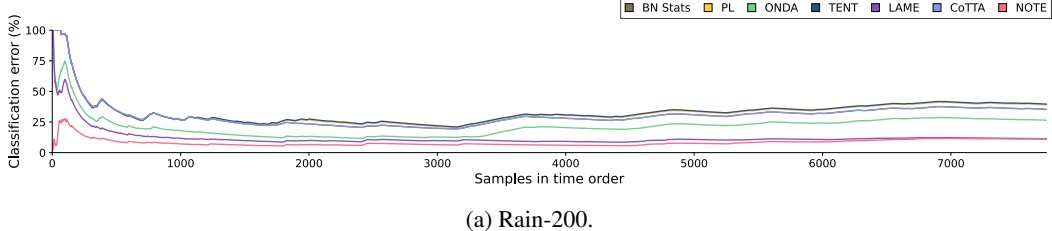

(a) Rain-200.

Figure 6: Illustration of the real-time cumulative classification error change of different methods on the KITTI dataset. The x-axis denotes the samples in order, whereas the y-axis denotes the error rate in percentage. Note that some lines are not clearly visible due to overlap.

100 epochs and cosine annealing learning rate scheduling [26] with an initial learning rate of 0.1 and a batch size of 64.

**Error on the source domain**. We also measure the domain gap between the source and the targets in the three real-distribution datasets: Table 1 for KITTI, Table 2 for HARTH, and Table 3 for Extrasensory. As shown, there is a clear performance degradation from the source domain to the target domain. For HARTH and ExtraSensory, the performance degradation was severe (30∼40%p increased error rates compared with Source), indicating the importance of overcoming the domain shift problem in sensory applications.

Table 1: Average classification error (%) and their corresponding standard deviations on the KITTI dataset of the source model. **Bold** fonts indicate the lowest classification errors. Averaged over three runs.

| Method | Src domain | Rain | Avg |
|---|---|---|---|
| Source | **7.4 ± 1.0** | 12.3 ± 2.3 | 9.9 |

Table 2: Average classification error (%) and their corresponding standard deviations on the HARTH dataset of the source model. **Bold** fonts indicate the lowest classification errors. Averaged over three runs.

| Method | Src domain | S008 | S018 | S019 | S021 | S022 | S028 | S029 | Avg |
|---|---|---|---|---|---|---|---|---|---|
| Source | **11.7 ± 0.7** | 86.2 ± 1.3 | 44.7 ± 2.1 | 50.4 ± 9.5 | 74.8 ± 3.8 | 72.0 ± 2.6 | 53.0 ± 24.0 | 57.0 ± 16.7 | 56.2 |

Table 3: Average classification error (%) and their corresponding standard deviations on the ExtraSensory dataset of the source model. **Bold** fonts indicate the lowest classification errors. Averaged over three runs.

| Method | Src domain | 4FC | 598 | 619 | 797 | A5C | C48 | D7D | Avg |
|---|---|---|---|---|---|---|---|---|---|
| Source | **8.3 ± 0.7** | 34.6 ± 2.5 | 40.1 ± 0.7 | 63.8 ± 5.7 | 45.3 ± 2.4 | 64.6 ± 3.7 | 39.6 ± 6.8 | 63.0 ± 3.9 | 44.9 |

# B Domain-wise results

## B.1 Robustness to corruptions

Table 4: Average classification error (%) and their corresponding standard deviations on CIFAR10-C with **temporally correlated test streams**, shown per corruption. **Bold** fonts indicate the lowest classification errors, while Red fonts show performance degradation after adaptation. Averaged over three runs.

| Method | Gaussian | Shot | Impulse | Defocus | Glass | Motion | Zoom | Snow | Frost | Fog | Brightness | Contrast | Elastic | Pixelate | JPEG | Avg |
|---|---|---|---|---|---|---|---|---|---|---|---|---|---|---|---|---|
| Source | 74.0 ± 3.3 | 66.8 ± 3.5 | 75.3 ± 4.2 | 43.3 ± 2.7 | 48.0 ± 2.7 | 32.6 ± 1.2 | 35.2 ± 2.6 | 22.0 ± 0.4 | 33.0 ± 2.5 | 25.9 ± 0.9 | 8.5 ± 0.3 | 66.1 ± 1.8 | 23.4 ± 0.7 | 53.6 ± 0.7 | 26.8 ± 0.7 | 42.3 |
| BN Stats [29] | 77.2 ± 0.7 | 76.7 ± 1.0 | 78.9 ± 0.8 | 70.0 ± 1.7 | 78.6 ± 0.6 | 70.5 ± 1.5 | 71.1 ± 1.4 | 72.5 ± 1.4 | 71.9 ± 1.1 | 70.6 ± 1.6 | 68.7 ± 1.9 | 69.1 ± 1.9 | 75.1 ± 1.5 | 73.6 ± 1.4 | 76.8 ± 1.4 | 73.4 |
| ONDA [27] | 69.3 ± 1.0 | 68.5 ± 1.0 | 71.8 ± 0.6 | 58.5 ± 1.4 | 71.0 ± 0.2 | 59.9 ± 1.0 | 59.5 ± 1.0 | 62.4 ± 1.4 | 62.1 ± 1.0 | 59.6 ± 1.3 | 55.6 ± 1.4 | 58.4 ± 1.4 | 65.6 ± 1.0 | 63.9 ± 1.4 | 67.6 ± 1.1 | 63.6 |
| PL [22] | 78.3 ± 1.0 | 78.0 ± 1.5 | 80.4 ± 1.0 | 72.2 ± 1.6 | 80.1 ± 1.2 | 72.4 ± 2.2 | 73.1 ± 1.4 | 74.5 ± 2.5 | 73.9 ± 1.8 | 73.4 ± 1.7 | 71.5 ± 2.7 | 71.7 ± 2.5 | 77.3 ± 2.1 | 75.7 ± 1.5 | 78.6 ± 2.7 | 75.4 |
| TENT [41] | 79.0 ± 2.9 | 78.8 ± 2.8 | 80.6 ± 2.2 | 73.3 ± 1.7 | 80.5 ± 2.9 | 74.4 ± 2.4 | 74.5 ± 3.3 | 74.8 ± 2.2 | 75.0 ± 2.3 | 74.0 ± 2.2 | 72.3 ± 3.4 | 74.9 ± 3.2 | 78.2 ± 2.8 | 76.5 ± 2.9 | 79.0 ± 2.9 | 76.4 |
| LAME [4] | 73.6 ± 5.2 | 64.8 ± 4.6 | 74.8 ± 6.4 | 36.2 ± 4.4 | 37.7 ± 5.3 | 24.9 ± 1.6 | 27.9 ± 3.4 | **12.4** ± 1.0 | 22.4 ± 3.9 | 19.4 ± 0.9 | **3.6** ± 0.3 | 65.1 ± 1.5 | **12.6** ± 0.8 | 50.3 ± 0.9 | **16.4** ± 1.2 | 36.2 |
| CoTTA [44] | 77.0 ± 0.7 | 76.8 ± 0.6 | 79.0 ± 0.7 | 74.1 ± 0.9 | 79.6 ± 0.6 | 74.3 ± 0.5 | 74.0 ± 0.8 | 74.8 ± 1.1 | 73.3 ± 0.9 | 72.9 ± 0.5 | 72.2 ± 0.9 | 76.5 ± 0.8 | 76.5 ± 0.9 | 75.1 ± 0.8 | 76.6 ± 0.6 | 75.5 |
| NOTE | **34.9** ± 1.6 | **32.3** ± 3.1 | **39.6** ± 2.5 | **13.6** ± 0.5 | **35.8** ± 1.9 | **11.8** ± 0.8 | **14.5** ± 0.5 | 14.1 ± 0.6 | **15.2** ± 1.3 | **14.2** ± 0.6 | 7.7 ± 0.3 | **7.6** ± 0.6 | 20.8 ± 0.7 | **27.7** ± 2.6 | 26.4 ± 0.5 | **21.1** |

Table 5: Average classification error (%) and their corresponding standard deviations on CIFAR100-C with **temporally correlated test streams**, shown per corruption. **Bold** fonts indicate the lowest classification errors, while Red fonts show performance degradation after adaptation. Averaged over three runs.

| Method | Gaussian | Shot | Impulse | Defocus | Glass | Motion | Zoom | Snow | Frost | Fog | Brightness | Contrast | Elastic | Pixelate | JPEG | Avg |
|---|---|---|---|---|---|---|---|---|---|---|---|---|---|---|---|---|
| Source | 88.1 ± 0.2 | 86.8 ± 0.6 | 93.7 ± 0.6 | 64.9 ± 0.4 | 79.7 ± 0.9 | 55.5 ± 0.3 | 57.7 ± 0.2 | 53.8 ± 0.4 | 66.3 ± 0.8 | 59.3 ± 0.4 | 33.0 ± 1.1 | 81.4 ± 0.4 | 49.2 ± 0.4 | 73.6 ± 1.1 | 55.5 ± 0.3 | 66.6 |
| BN Stats [29] | 73.9 ± 0.5 | 73.5 ± 0.4 | 77.2 ± 0.7 | 56.9 ± 0.2 | 72.3 ± 0.5 | 58.8 ± 0.3 | 57.9 ± 0.4 | 65.3 ± 0.4 | 65.0 ± 0.4 | 62.4 ± 0.6 | 55.6 ± 0.2 | 57.6 ± 0.4 | 64.6 ± 0.5 | 63.6 ± 0.3 | 71.0 ± 0.4 | 65.0 |
| ONDA [27] | **63.0** ± 0.7 | **62.5** ± 0.4 | **68.0** ± 0.5 | 37.3 ± 0.2 | **60.0** ± 0.2 | 40.0 ± 0.3 | 38.3 ± 0.1 | 49.6 ± 0.3 | 50.0 ± 0.6 | 45.2 ± 0.6 | 35.7 ± 0.2 | 40.9 ± 0.5 | 48.6 ± 0.5 | **46.9** ± 0.3 | 57.5 ± 0.2 | 49.6 |
| PL [22] | 71.9 ± 1.4 | 72.0 ± 0.5 | 76.3 ± 0.7 | 59.3 ± 0.8 | 73.8 ± 0.9 | 61.5 ± 0.9 | 59.9 ± 0.5 | 67.1 ± 0.9 | 66.7 ± 1.4 | 63.0 ± 1.0 | 57.9 ± 0.5 | 62.2 ± 1.5 | 67.6 ± 1.0 | 65.2 ± 0.3 | 71.1 ± 0.5 | 66.4 |
| TENT [41] | 71.8 ± 0.9 | 71.0 ± 0.4 | 76.4 ± 1.2 | 60.2 ± 0.6 | 75.0 ± 1.0 | 61.9 ± 0.9 | 60.2 ± 0.7 | 67.8 ± 0.5 | 67.8 ± 0.7 | 63.3 ± 1.1 | 58.4 ± 0.7 | 65.0 ± 1.8 | 68.4 ± 0.9 | 65.0 ± 0.2 | 71.8 ± 0.1 | 66.9 |
| LAME [4] | 89.0 ± 1.1 | 87.1 ± 0.8 | 94.5 ± 0.7 | 62.3 ± 1.2 | 79.7 ± 1.2 | 49.4 ± 1.0 | 52.8 ± 0.3 | 46.6 ± 0.4 | 63.9 ± 1.9 | 55.6 ± 1.2 | 25.2 ± 0.6 | 82.4 ± 0.2 | **40.8** ± 0.5 | 71.9 ± 1.4 | **47.8** ± 0.7 | 63.3 |
| CoTTA [44] | 68.6 ± 0.3 | 67.9 ± 0.4 | 71.4 ± 0.4 | 60.7 ± 0.4 | 69.9 ± 0.4 | 60.8 ± 0.5 | 60.2 ± 0.2 | 64.0 ± 0.3 | 62.9 ± 0.5 | 63.2 ± 0.6 | 56.7 ± 0.2 | 65.6 ± 0.3 | 64.5 ± 0.3 | 60.9 ± 0.0 | 65.3 ± 0.1 | 64.2 |
| NOTE | 66.2 ± 0.8 | 64.2 ± 1.6 | 72.6 ± 0.4 | **37.2** ± 0.8 | 61.1 ± 0.7 | **35.4** ± 0.3 | **37.4** ± 0.4 | **40.0** ± 0.4 | **42.5** ± 0.3 | **43.4** ± 0.5 | 29.4 ± 0.1 | **32.1** ± 0.5 | 44.3 ± 0.4 | 47.5 ± 0.6 | 51.3 ± 0.3 | **47.0** |

Table 6: Average classification error (%) and their corresponding standard deviations on ImageNet-C with **temporally correlated test streams**, shown per corruption. **Bold** fonts indicate the lowest classification errors, while Red fonts show performance degradation after adaptation. Averaged over three runs.

| Method | Gaussian | Shot | Impulse | Defocus | Glass | Motion | Zoom | Snow | Frost | Fog | Brightness | Contrast | Elastic | Pixelate | JPEG | Avg |
|---|---|---|---|---|---|---|---|---|---|---|---|---|---|---|---|---|
| Source | 98.4 | 97.7 | 98.4 | 90.6 | 92.5 | 89.8 | 81.8 | 89.5 | 85.0 | 86.4 | 51.1 | 97.2 | 85.3 | 76.9 | 71.7 | 86.1 |
| | ± 0.0 | ± 0.0 | ± 0.0 | ± 0.0 | ± 0.0 | ± 0.0 | ± 0.0 | ± 0.0 | ± 0.0 | ± 0.0 | ± 0.0 | ± 0.0 | ± 0.0 | ± 0.0 | ± 0.0 | |
| BN Stats | 98.3 | 98.1 | 98.4 | 98.7 | 98.8 | 97.8 | 96.6 | 96.2 | 96.0 | 95.1 | 93.1 | 98.6 | 96.3 | 95.6 | 96.1 | 96.9 |
| | ± 0.0 | ± 0.0 | ± 0.0 | ± 0.0 | ± 0.0 | ± 0.0 | ± 0.0 | ± 0.0 | ± 0.0 | ± 0.0 | ± 0.0 | ± 0.0 | ± 0.0 | ± 0.0 | ± 0.0 | |
| ONDA | 95.1 | 94.7 | 95.0 | 96.2 | 96.1 | 92.5 | 87.2 | 87.4 | 87.8 | 82.7 | 71.0 | 96.4 | 84.9 | 81.7 | 86.1 | 89.0 |
| | ± 0.0 | ± 0.0 | ± 0.0 | ± 0.0 | ± 0.0 | ± 0.0 | ± 0.0 | ± 0.0 | ± 0.0 | ± 0.0 | ± 0.0 | ± 0.0 | ± 0.0 | ± 0.0 | ± 0.0 | |
| PL | 99.3 | 99.3 | 99.4 | 99.5 | 99.4 | 99.5 | 98.8 | 99.1 | 99.2 | 98.1 | 97.3 | 99.8 | 98.4 | 98.5 | 98.5 | 98.9 |
| | ± 0.0 | ± 0.0 | ± 0.0 | ± 0.0 | ± 0.0 | ± 0.0 | ± 0.0 | ± 0.0 | ± 0.0 | ± 0.0 | ± 0.1 | ± 0.0 | ± 0.0 | ± 0.0 | ± 0.0 | |
| TENT | 98.3 | 98.1 | 98.4 | 98.7 | 98.8 | 97.8 | 96.6 | 96.2 | 96.0 | 95.1 | 93.1 | 98.6 | 96.3 | 95.6 | 96.1 | 96.9 |
| | ± 0.0 | ± 0.0 | ± 0.0 | ± 0.0 | ± 0.0 | ± 0.0 | ± 0.0 | ± 0.0 | ± 0.0 | ± 0.0 | ± 0.0 | ± 0.0 | ± 0.0 | ± 0.0 | ± 0.0 | |
| LAME | 98.1 | 97.1 | 98.0 | **87.9** | **90.9** | 87.1 | **78.3** | 87.1 | 80.2 | 81.5 | **39.8** | 96.4 | 82.5 | 70.7 | **64.9** | 82.7 |
| | ± 0.0 | ± 0.0 | ± 0.0 | **± 0.0** | **± 0.0** | ± 0.0 | **± 0.0** | ± 0.0 | ± 0.0 | ± 0.0 | **± 0.0** | ± 0.0 | ± 0.0 | ± 0.0 | **± 0.0** | |
| CoTTA | 98.2 | 98.1 | 98.3 | 98.8 | 98.8 | 97.7 | 96.8 | 96.6 | 96.3 | 95.3 | 93.5 | 98.8 | 96.5 | 95.6 | 96.2 | 97.0 |
| | ± 0.0 | ± 0.0 | ± 0.0 | ± 0.0 | ± 0.0 | ± 0.0 | ± 0.0 | ± 0.1 | ± 0.0 | ± 0.0 | ± 0.0 | ± 0.0 | ± 0.0 | ± 0.0 | ± 0.0 | |
| NOTE | **94.7** | **93.7** | **94.5** | 91.2 | 91.0 | **83.3** | 79.0 | **79.0** | **78.7** | **66.3** | 48.0 | **94.1** | 76.9 | 62.6 | 76.6 | **80.6** |
| | **± 0.1** | **± 0.3** | **± 0.1** | ± 0.1 | ± 0.2 | **± 0.1** | ± 0.2 | **± 0.4** | **± 0.3** | **± 0.6** | ± 0.4 | **± 0.1** | ± 0.6 | ± 0.7 | ± 0.6 | |

Table 7: Average classification error (%) and their corresponding standard deviations on MNIST-C with **temporally correlated test streams**, shown per corruption. **Bold** fonts indicate the lowest classification errors, while Red fonts show performance degradation after adaptation. Averaged over three runs.

| Method | Shot | Impulse | Glass | Motion | Shear | Scale | Rotate | Brightness | Translate | Stripe | Fog | Spatter | Dotted line | Zigzag | Canny edges | Avg |
|---|---|---|---|---|---|---|---|---|---|---|---|---|---|---|---|---|
| Source | 3.7 | 27.3 | 20.4 | 4.6 | 2.2 | 5.1 | 6.5 | 21.1 | 13.8 | 17.4 | 66.6 | 3.8 | 3.7 | 18.2 | 26.4 | 16.1 |
| | ± 0.7 | ± 5.5 | ± 6.4 | ± 0.5 | ± 0.5 | ± 1.0 | ± 1.0 | ± 22.9 | ± 1.4 | ± 17.0 | ± 14.7 | ± 0.4 | ± 0.4 | ± 3.0 | ± 11.4 | |
| BN Stats [29] | 72.0 | 75.2 | 73.7 | 72.1 | 71.2 | 71.4 | 71.2 | 71.6 | 78.5 | 72.3 | 70.8 | 71.6 | 73.8 | 74.6 | 72.3 | 72.8 |
| | ± 0.6 | ± 0.8 | ± 1.0 | ± 0.8 | ± 1.1 | ± 0.6 | ± 0.3 | ± 0.6 | ± 0.2 | ± 1.2 | ± 1.2 | ± 0.9 | ± 0.7 | ± 0.6 | ± 0.3 | |
| ONDA [27] | 53.3 | 59.9 | 59.2 | 54.1 | 51.6 | 53.9 | 54.6 | 50.5 | 65.2 | 57.5 | 54.8 | 54.2 | 55.4 | 61.0 | 56.7 | 56.1 |
| | ± 3.0 | ± 3.0 | ± 3.3 | ± 3.5 | ± 2.2 | ± 2.5 | ± 2.0 | ± 2.3 | ± 2.1 | ± 0.7 | ± 2.9 | ± 3.0 | ± 2.8 | ± 2.2 | ± 2.1 | |
| PL [22] | 73.7 | 76.4 | 75.3 | 74.7 | 72.7 | 73.3 | 73.7 | 73.7 | 78.7 | 74.1 | 75.8 | 72.5 | 75.8 | 76.9 | 74.5 | 74.8 |
| | ± 1.0 | ± 0.4 | ± 0.5 | ± 1.1 | ± 0.9 | ± 1.6 | ± 0.9 | ± 1.0 | ± 0.3 | ± 1.4 | ± 2.6 | ± 0.8 | ± 0.6 | ± 1.4 | ± 0.1 | |
| TENT [41] | 74.7 | 78.1 | 76.6 | 76.1 | 75.8 | 73.7 | 75.2 | 75.4 | 78.9 | 76.7 | 81.4 | 73.9 | 77.3 | 79.2 | 75.8 | 76.6 |
| | ± 1.1 | ± 0.9 | ± 0.6 | ± 0.7 | ± 1.1 | ± 1.3 | ± 1.1 | ± 0.3 | ± 0.2 | ± 1.8 | ± 1.7 | ± 0.5 | ± 0.7 | ± 2.0 | ± 1.0 | |
| LAME [4] | **1.1** | 17.0 | **12.5** | **1.1** | **0.4** | **1.5** | **2.3** | 17.2 | **6.0** | **12.3** | 68.3 | **0.7** | **0.7** | 13.2 | 22.1 | 11.8 |
| | **± 0.3** | ± 8.7 | **± 6.5** | **± 0.3** | **± 0.2** | **± 0.6** | **± 0.6** | ± 26.0 | **± 2.3** | **± 17.2** | ± 15.8 | **± 0.3** | **± 0.4** | ± 3.4 | ± 12.3 | |
| CoTTA [44] | 76.9 | 79.4 | 79.1 | 77.6 | 75.4 | 76.2 | 77.6 | 76.0 | 81.6 | 76.8 | 78.0 | 77.6 | 79.3 | 80.6 | 77.6 | 78.0 |
| | ± 0.5 | ± 0.4 | ± 0.5 | ± 0.6 | ± 0.4 | ± 1.3 | ± 0.2 | ± 0.5 | ± 0.9 | ± 0.6 | ± 0.4 | ± 0.6 | ± 0.4 | ± 1.0 | ± 0.5 | |
| NOTE | 3.9 | **13.8** | 14.3 | 3.3 | 1.7 | 3.8 | 6.5 | **0.9** | 8.0 | 14.4 | **1.6** | 3.9 | 4.5 | **12.6** | **13.4** | **7.1** |
| | ± 1.3 | **± 2.4** | ± 1.5 | ± 2.4 | ± 0.2 | ± 0.7 | ± 0.3 | **± 0.0** | ± 1.2 | ± 8.1 | **± 0.3** | ± 0.4 | ± 1.2 | **± 2.5** | **± 3.9** | |

Table 8: Average classification error (%) and their corresponding standard deviations on CIFAR10-C with **uniformly distributed test streams**, shown per domain. **Bold** fonts indicate the lowest classification errors, while Red fonts show performance degradation after adaptation. NOTE* indicates NOTE used directly with test batches (without using PBRS). Averaged over three runs.

| Method | Gaussian | Shot | Impulse | Defocus | Glass | Motion | Zoom | Snow | Frost | Fog | Brightness | Contrast | Elastic | Pixelate | JPEG | Avg |
|---|---|---|---|---|---|---|---|---|---|---|---|---|---|---|---|---|
| Source | 74.0 ± 3.3 | 66.8 ± 3.5 | 75.3 ± 4.2 | 43.3 ± 2.7 | 48.0 ± 2.7 | 32.6 ± 1.2 | 35.2 ± 2.6 | 22.0 ± 0.4 | 33.0 ± 2.5 | 25.9 ± 0.9 | 8.5 ± 0.3 | 66.1 ± 1.8 | 23.4 ± 0.7 | 53.6 ± 0.7 | 26.8 ± 0.7 | 42.3 |
| BN Stats [29] | 33.1 ± 0.9 | 31.1 ± 1.0 | 39.8 ± 0.9 | 12.3 ± 0.4 | 34.8 ± 0.3 | 13.7 ± 0.3 | 12.6 ± 0.4 | 18.3 ± 0.7 | 19.9 ± 0.6 | 14.5 ± 0.6 | 9.3 ± 0.3 | 13.0 ± 0.3 | 23.3 ± 0.3 | 20.8 ± 0.2 | 28.0 ± 0.6 | 21.6 |
| ONDA [27] | 33.4 ± 0.6 | 31.3 ± 0.9 | 40.0 ± 1.1 | 12.3 ± 0.4 | 34.6 ± 0.7 | 13.7 ± 0.3 | 12.4 ± 0.5 | 18.3 ± 0.6 | 19.8 ± 0.8 | 14.3 ± 0.4 | 9.1 ± 0.0 | 14.0 ± 0.2 | 23.3 ± 0.4 | 20.9 ± 0.2 | 28.0 ± 0.7 | 21.7 |
| PL [22] | 29.4 ± 1.1 | 26.3 ± 1.0 | 36.8 ± 1.6 | 13.7 ± 0.4 | 36.5 ± 1.1 | 14.0 ± 1.0 | 13.5 ± 0.2 | 19.7 ± 0.8 | 21.2 ± 0.6 | 15.6 ± 1.5 | 10.0 ± 0.6 | 14.8 ± 0.2 | 24.5 ± 2.0 | 20.1 ± 0.9 | 27.4 ± 1.3 | 21.6 |
| TENT [41] | 25.3 ± 0.8 | 23.1 ± 1.1 | 32.1 ± 1.2 | **11.7 ± 0.6** | 33.1 ± 3.0 | 13.2 ± 1.1 | **11.2 ± 0.1** | 15.9 ± 0.3 | 18.8 ± 0.7 | **12.9 ± 0.8** | 8.6 ± 0.3 | 14.4 ± 0.6 | 21.7 ± 0.9 | **16.5 ± 0.8** | 23.6 ± 0.7 | 18.8 |
| LAME [4] | 78.2 ± 3.6 | 70.6 ± 4.0 | 80.5 ± 4.5 | 46.6 ± 1.9 | 48.0 ± 3.8 | 34.2 ± 0.4 | 37.4 ± 1.5 | 20.8 ± 0.8 | 30.5 ± 4.1 | 26.9 ± 1.8 | 9.8 ± 0.2 | 71.9 ± 1.0 | 24.2 ± 0.9 | 56.4 ± 0.8 | 25.8 ± 0.9 | 44.1 |
| CoTTA [44] | **23.1 ± 0.7** | **21.5 ± 0.6** | **28.0 ± 0.3** | **11.7 ± 0.5** | **29.2 ± 0.6** | 13.3 ± 0.6 | 12.0 ± 0.5 | 16.6 ± 0.2 | 16.6 ± 0.3 | 13.8 ± 0.4 | 8.8 ± 0.2 | 14.9 ± 0.5 | **20.6 ± 0.7** | 17.3 ± 0.5 | **19.9 ± 0.4** | 17.8 |
| NOTE | 33.5 ± 1.7 | 30.0 ± 1.6 | 38.2 ± 0.9 | 12.6 ± 0.8 | 34.4 ± 0.8 | **11.5 ± 0.5** | 12.9 ± 0.6 | **14.1 ± 0.2** | 15.2 ± 0.8 | 14.0 ± 0.6 | **7.4 ± 0.2** | 7.8 ± 0.2 | 20.7 ± 0.3 | 24.7 ± 0.4 | 24.2 ± 0.4 | 20.1 |
| NOTE* | 23.8 ± 0.7 | 23.0 ± 0.9 | 31.1 ± 0.3 | 11.8 ± 0.6 | 30.9 ± 1.3 | 11.8 ± 0.4 | 11.9 ± 0.7 | 15.3 ± 1.3 | **14.0 ± 0.7** | 13.3 ± 0.7 | 8.6 ± 0.2 | **7.5 ± 0.3** | 21.2 ± 0.3 | 16.9 ± 0.6 | 23.0 ± 1.2 | **17.6** |

Table 9: Average classification error (%) and their corresponding standard deviations on CIFAR100-C with **uniformly distributed test streams**, shown per domain. **Bold** fonts indicate the lowest classification errors, while Red fonts show performance degradation after adaptation. Averaged over three runs. NOTE* indicates NOTE used directly with test batches (without using PBRS)

| Method | Gaussian | Shot | Impulse | Defocus | Glass | Motion | Zoom | Snow | Frost | Fog | Brightness | Contrast | Elastic | Pixelate | JPEG | Avg |
|---|---|---|---|---|---|---|---|---|---|---|---|---|---|---|---|---|
| Source | 88.1 ± 0.2 | 86.8 ± 0.6 | 93.7 ± 0.6 | 64.9 ± 0.4 | 79.7 ± 0.9 | 55.5 ± 0.3 | 57.7 ± 0.2 | 53.8 ± 0.4 | 66.3 ± 0.8 | 59.3 ± 0.4 | 33.0 ± 0.3 | 81.4 ± 0.4 | 49.2 ± 0.4 | 73.6 ± 1.1 | 55.5 ± 0.3 | 66.6 |
| BN Stats [29] | 60.9 ± 0.8 | 59.9 ± 0.6 | 65.7 ± 0.8 | 33.7 ± 0.4 | 57.6 ± 0.4 | 36.5 ± 0.4 | 35.2 ± 0.4 | 46.7 ± 0.3 | 46.9 ± 0.4 | 42.8 ± 0.7 | 32.3 ± 0.4 | 35.6 ± 0.5 | 45.8 ± 0.3 | 43.6 ± 0.3 | 55.5 ± 0.2 | 46.6 |
| ONDA [27] | 60.8 ± 0.9 | 60.2 ± 0.5 | 66.0 ± 0.6 | 33.9 ± 0.4 | 57.5 ± 0.4 | 36.3 ± 0.4 | 34.6 ± 0.4 | 46.5 ± 0.3 | 47.2 ± 0.3 | 42.1 ± 0.6 | 32.1 ± 0.5 | 36.4 ± 0.4 | 45.5 ± 0.1 | 43.4 ± 0.8 | 55.1 ± 0.1 | 46.5 |
| PL [22] | 52.2 ± 0.9 | 50.3 ± 1.0 | 59.4 ± 0.9 | 33.5 ± 0.5 | 54.0 ± 0.6 | 35.7 ± 0.3 | 33.1 ± 0.5 | 42.8 ± 0.9 | 44.5 ± 1.6 | 39.2 ± 1.3 | 30.9 ± 0.2 | 35.5 ± 0.2 | 45.5 ± 1.0 | 39.9 ± 0.3 | 50.4 ± 1.3 | 43.1 |
| TENT [41] | **48.7 ± 0.8** | **47.2 ± 0.6** | **55.6 ± 0.9** | **31.5 ± 0.2** | **50.9 ± 0.5** | 33.5 ± 0.4 | **31.7 ± 0.2** | 39.6 ± 0.3 | 41.0 ± 0.1 | 36.8 ± 0.7 | 29.4 ± 0.3 | 33.6 ± 0.4 | **42.3 ± 0.6** | **36.8 ± 0.5** | 46.4 ± 0.5 | **40.3** |
| LAME [4] | 91.0 ± 1.0 | 89.5 ± 1.0 | 95.2 ± 0.7 | 68.1 ± 0.9 | 82.7 ± 1.1 | 57.1 ± 0.5 | 60.2 ± 0.3 | 54.7 ± 0.3 | 68.9 ± 1.2 | 61.8 ± 0.6 | 33.7 ± 0.5 | 85.2 ± 0.4 | 50.3 ± 0.2 | 76.7 ± 1.3 | 56.2 ± 0.5 | 68.8 |
| CoTTA [44] | 52.8 ± 0.7 | 51.0 ± 0.4 | 56.9 ± 0.6 | 35.8 ± 0.4 | 53.9 ± 0.2 | 37.9 ± 0.4 | 36.8 ± 0.1 | 45.2 ± 0.5 | 44.5 ± 0.1 | 44.0 ± 0.2 | 32.2 ± 0.5 | 41.3 ± 1.4 | 46.1 ± 0.1 | 39.7 ± 0.3 | 46.9 ± 0.7 | 44.3 |
| NOTE | 65.6 ± 1.0 | 62.6 ± 0.7 | 72.0 ± 0.2 | 36.8 ± 0.7 | 60.5 ± 0.7 | 34.9 ± 0.5 | 36.7 ± 0.2 | 39.6 ± 0.2 | 41.7 ± 0.2 | 42.3 ± 0.6 | **28.6 ± 0.2** | 32.3 ± 0.9 | 43.8 ± 0.2 | 47.7 ± 0.4 | 50.9 ± 0.2 | 46.4 |
| NOTE* | 51.8 ± 1.0 | 50.0 ± 0.3 | 60.7 ± 0.4 | 32.6 ± 0.2 | 54.4 ± 0.3 | **33.0 ± 0.2** | 33.5 ± 0.4 | **38.5 ± 0.3** | **38.6 ± 0.1** | **36.7 ± 0.3** | 29.7 ± 0.5 | **27.3 ± 0.3** | 43.2 ± 0.4 | 37.1 ± 0.2 | 47.6 ± 0.9 | 41.0 |

Table 10: Average classification error (%) and their corresponding standard deviations on ImageNet-C with **temporally correlated test streams**, shown per corruption. **Bold** fonts indicate the lowest classification errors, while Red fonts show performance degradation after adaptation. Averaged over three runs.

| Method | Gaussian | Shot | Impulse | Defocus | Glass | Motion | Zoom | Snow | Frost | Fog | Brightness | Contrast | Elastic | Pixelate | JPEG | Avg |
|---|---|---|---|---|---|---|---|---|---|---|---|---|---|---|---|---|
| Source | 98.4 ±0.0 | 97.7 ±0.0 | 98.4 ±0.0 | 90.6 ±0.0 | 92.5 ±0.0 | 89.8 ±0.0 | 81.8 ±0.0 | 89.5 ±0.0 | 85.0 ±0.0 | 86.4 ±0.0 | 51.1 ±0.0 | 97.2 ±0.0 | 85.3 ±0.0 | 76.9 ±0.0 | 71.7 ±0.0 | 86.1 |
| BN Stats | 89.4 ±0.0 | 88.5 ±0.1 | 89.2 ±0.2 | 90.8 ±0.0 | 90.0 ±0.0 | 81.3 ±0.0 | 69.8 ±0.2 | 72.6 ±0.1 | 73.8 ±0.0 | 62.6 ±0.0 | 44.3 ±0.3 | 92.1 ±0.0 | 64.5 ±0.1 | 60.3 ±0.1 | 70.7 ±0.0 | 76.0 |
| ONDA | 89.2 ±0.0 | 88.2 ±0.0 | 89.0 ±0.1 | 90.9 ±0.1 | 90.0 ±0.1 | 81.6 ±0.1 | 69.5 ±0.0 | 72.6 ±0.1 | 73.7 ±0.0 | 62.7 ±0.1 | 43.9 ±0.0 | 92.1 ±0.0 | 64.3 ±0.0 | 60.1 ±0.1 | 70.0 ±0.0 | 75.9 |
| PL | 89.8 ±1.9 | 86.1 ±0.9 | 88.5 ±1.6 | 93.0 ±1.1 | 92.5 ±0.6 | 82.2 ±0.0 | 64.6 ±0.3 | 70.2 ±0.6 | 79.7 ±0.4 | **55.8** **±0.2** | 43.9 ±0.1 | 97.2 ±0.5 | **57.8** **±0.1** | **52.7** **±0.2** | **60.5** **±0.1** | 74.4 |
| TENT | 91.1 ±2.4 | 89.7 ±1.6 | 91.0 ±2.5 | 93.1 ±3.2 | 92.2 ±3.2 | 84.7 ±4.9 | 72.4 ±3.5 | 73.3 ±1.1 | 78.7 ±6.9 | 59.8 ±4.0 | 44.5 ±0.5 | 95.2 ±4.3 | 61.6 ±4.3 | 56.4 ±5.6 | 67.4 ±4.7 | 76.5 |
| LAME | 98.6 ±0.0 | 97.8 ±0.0 | 98.6 ±0.0 | 90.7 ±0.0 | 92.6 ±0.0 | 89.9 ±0.0 | 81.9 ±0.0 | 89.8 ±0.0 | 85.0 ±0.0 | 86.5 ±0.0 | 51.1 ±0.0 | 97.3 ±0.0 | 85.6 ±0.0 | 77.0 ±0.0 | 71.7 ±0.0 | 86.3 |
| CoTTA | **85.7** **±0.2** | **84.6** **±0.1** | **85.4** **±0.0** | 87.8 ±0.3 | 86.4 ±0.2 | 74.6 ±0.0 | **64.2** **±0.2** | 67.9 ±0.0 | 69.7 ±0.2 | 56.1 ±0.1 | **42.7** **±0.0** | 88.5 ±0.8 | 60.0 ±0.0 | 54.2 ±0.1 | 64.9 ±0.1 | 71.5 |
| NOTE | 87.6 ±0.1 | 85.7 ±0.1 | 87.2 ±0.2 | **83.3** **±0.2** | **83.2** **±0.2** | **73.6** **±0.0** | 65.4 ±0.2 | **65.0** **±0.0** | **68.6** **±0.1** | 57.9 ±0.0 | 43.5 ±0.1 | **75.9** **±0.1** | 61.2 ±0.1 | 54.1 ±0.0 | 62.8 ±0.1 | **70.3** |
| NOTE* | 89.5 ±0.4 | 87.9 ±0.2 | 88.9 ±0.3 | 84.6 ±0.2 | 83.7 ±0.2 | 74.4 ±0.1 | 66.6 ±0.1 | 66.1 ±0.2 | 71.2 ±0.1 | 58.2 ±0.1 | 44.7 ±0.1 | 78.8 ±0.1 | 61.2 ±0.2 | 54.8 ±0.0 | 64.8 ±0.1 | 71.7 |

Table 11: Average classification error (%) and their corresponding standard deviations on MNIST-C with **uniformly distributed test streams**, shown per domain. **Bold** fonts indicate the lowest classification errors, while Red fonts show performance degradation after adaptation. Averaged over three runs. NOTE* indicates NOTE used directly with test batches (without using PBRS).

| Method | Shot | Impulse | Glass | Motion | Shear | Scale | Rotate | Brightness | Translate | Stripe | Fog | Spatter | Dotted line | Zigzag | Canny edges | Avg |
|---|---|---|---|---|---|---|---|---|---|---|---|---|---|---|---|---|
| Source | 3.7 ±0.7 | 27.3 ±5.5 | 20.4 ±6.4 | 4.6 ±0.5 | 2.2 ±0.5 | 5.1 ±1.0 | 6.5 ±1.0 | 21.1 ±22.9 | 13.8 ±1.4 | 17.4 ±17.0 | 66.6 ±14.7 | 3.8 ±0.4 | 3.7 ±0.4 | 18.2 ±3.0 | 26.4 ±11.4 | 16.1 |
| BN Stats [29] | 2.9 ±0.7 | 7.0 ±1.6 | 9.1 ±1.0 | 3.0 ±0.8 | 2.0 ±0.3 | 3.8 ±0.2 | 6.1 ±0.7 | 1.1 ±0.1 | 12.5 ±0.8 | 6.5 ±2.6 | 2.2 ±0.5 | 3.3 ±0.3 | 2.5 ±0.2 | 11.4 ±0.2 | 6.7 ±0.9 | 5.3 |
| ONDA [27] | 2.6 ±0.6 | 6.5 ±1.4 | 8.6 ±1.0 | 2.8 ±0.8 | 1.8 ±0.2 | 3.5 ±0.2 | 5.7 ±0.7 | 1.0 ±0.1 | 11.7 ±1.1 | 6.1 ±2.6 | 2.6 ±0.9 | 3.0 ±0.4 | 2.2 ±0.2 | 11.0 ±0.4 | 6.2 ±0.8 | 5.0 |
| PL [22] | 1.6 ±0.3 | 3.5 ±0.7 | 4.8 ±0.8 | 1.7 ±0.0 | 1.5 ±0.0 | 2.3 ±0.1 | 4.9 ±0.7 | 0.8 ±0.1 | 6.8 ±0.8 | 2.7 ±0.6 | 1.0 ±0.0 | 2.2 ±0.3 | 1.7 ±0.2 | 5.3 ±0.4 | 3.9 ±0.9 | 3.0 |
| TENT [41] | 1.4 ±0.1 | 2.8 ±0.4 | **3.8** **±0.5** | 1.5 ±0.0 | 1.2 ±0.0 | 1.8 ±0.1 | 3.6 ±0.2 | **0.7** **±0.1** | 4.6 ±0.7 | **1.9** **±0.2** | 0.8 ±0.0 | **1.7** **±0.1** | **1.3** **±0.1** | **4.5** **±0.6** | **3.1** **±0.5** | 2.3 |
| LAME [4] | 3.0 ±0.8 | 30.7 ±8.3 | 18.9 ±5.8 | 3.4 ±0.5 | 1.9 ±0.3 | 4.2 ±0.5 | 6.3 ±0.9 | 25.9 ±29.8 | 13.9 ±1.9 | 18.5 ±21.2 | 78.2 ±9.8 | 3.3 ±0.7 | 3.2 ±0.3 | 19.3 ±3.2 | 28.0 ±12.7 | 17.2 |
| CoTTA [44] | 2.6 ±0.6 | 6.6 ±1.7 | 8.7 ±0.9 | 2.7 ±0.7 | 1.8 ±0.3 | 3.2 ±0.0 | 5.6 ±0.8 | 1.0 ±0.1 | 14.3 ±1.1 | 7.7 ±6.0 | 1.9 ±0.5 | 2.9 ±0.3 | 2.2 ±0.1 | 13.6 ±1.4 | 6.1 ±0.6 | 5.4 |
| NOTE | 2.5 ±0.8 | 10.7 ±1.9 | 10.9 ±2.0 | 2.0 ±0.3 | 1.5 ±0.0 | 2.4 ±0.1 | 5.5 ±0.3 | 0.9 ±0.1 | 5.5 ±0.2 | 12.1 ±5.7 | 1.2 ±0.1 | 2.8 ±0.3 | 3.0 ±0.1 | 10.9 ±1.6 | 9.1 ±0.4 | 5.4 |
| NOTE* | **1.3** **±0.2** | **2.7** **±0.1** | **3.8** **±0.5** | **1.3** **±0.1** | **1.1** **±0.1** | **1.6** **±0.0** | **3.5** **±0.1** | **0.7** **±0.0** | **2.8** **±0.0** | 2.2 ±0.1 | **0.7** **±0.1** | **1.7** **±0.4** | 1.4 ±0.2 | 4.8 ±1.1 | 3.5 ±0.1 | **2.2** |

## B.2 Real distributions with domain shift

Since the adaptation is done from a single source domain to a single target domain in KITTI, no further per-domain tables are specified here.

Table 12: Average classification error (%) and their corresponding standard deviations on HARTH with **real test streams**, shown per domain. **Bold** fonts indicate the lowest classification errors, while Red fonts show performance degradation after adaptation. Averaged over three runs.

| Method | S008 | S018 | S019 | S021 | S022 | S028 | S029 | Avg |
|---|---|---|---|---|---|---|---|---|
| Source | 86.2 ± 1.3 | 44.7 ± 2.1 | 50.4 ± 9.5 | 74.8 ± 3.8 | 72.0 ± 2.6 | 53.0 ± 24.0 | 57.0 ± 16.7 | 62.6 |
| BN Stats [29] | 70.3 ± 1.4 | 73.8 ± 1.3 | 68.1 ± 3.0 | 64.9 ± 0.9 | 68.5 ± 0.3 | 65.5 ± 0.5 | 69.4 ± 1.4 | 68.6 |
| ONDA [27] | 75.3 ± 4.0 | 60.4 ± 0.9 | 63.1 ± 4.6 | 67.9 ± 0.4 | 70.0 ± 3.8 | 73.6 ± 0.7 | 74.5 ± 4.4 | 69.3 |
| PL [22] | 60.4 ± 1.3 | 71.4 ± 1.5 | 62.9 ± 1.9 | 61.8 ± 1.2 | 63.1 ± 0.4 | 64.5 ± 0.8 | 69.4 ± 2.0 | 64.8 |
| TENT [41] | **59.5 ± 0.3** | 71.0 ± 1.6 | 62.2 ± 1.9 | **61.1 ± 1.1** | **61.7 ± 0.4** | 64.1 ± 0.5 | 69.3 ± 2.1 | 64.1 |
| LAME [4] | 85.5 ± 1.7 | 43.4 ± 2.0 | 48.8 ± 10.9 | 73.2 ± 3.8 | 70.7 ± 2.6 | 51.2 ± 29.4 | 54.1 ± 20.6 | 61.0 |
| CoTTA [44] | 70.4 ± 1.4 | 73.8 ± 1.3 | 68.2 ± 2.9 | 64.9 ± 1.0 | 68.5 ± 0.2 | 65.5 ± 0.5 | 69.4 ± 1.4 | 68.7 |
| NOTE | 84.8 ± 0.7 | **32.9 ± 1.8** | **36.3 ± 10.9** | 69.1 ± 2.4 | 67.1 ± 1.2 | **30.0 ± 13.8** | **36.6 ± 9.8** | **51.0** |

Table 13: Average classification error (%) and their corresponding standard deviations on Extrasensory with **real test streams**, shown per domain. **Bold** fonts indicate the lowest classification errors, while Red fonts show performance degradation after adaptation. Due to the length of the name of each domain, denoted here with the first three characters. Averaged over three runs.

| Method | 4FC | 598 | 619 | 797 | A5D | C48 | D7D | Avg |
|---|---|---|---|---|---|---|---|---|
| Source | 34.6 ± 2.5 | 40.1 ± 0.7 | 63.8 ± 5.7 | 45.3 ± 2.4 | 64.6 ± 3.7 | 39.6 ± 6.8 | 63.0 ± 3.9 | 50.2 |
| BN Stats[29] | 61.7 ± 4.2 | 50.1 ± 5.1 | 51.6 ± 1.5 | 59.4 ± 1.1 | 54.4 ± 1.0 | 52.4 ± 2.8 | 62.6 ± 2.9 | 56.0 |
| ONDA [27] | 36.3 ± 3.5 | 44.0 ± 2.2 | **50.8 ± 2.4** | 56.1 ± 1.9 | 59.7 ± 2.7 | 43.5 ± 5.9 | **46.7 ± 4.2** | 48.2 |
| PL [22] | 62.2 ± 4.3 | 50.0 ± 5.1 | 51.7 ± 1.8 | 59.2 ± 1.1 | 53.9 ± 1.1 | 52.3 ± 2.9 | 62.8 ± 3.0 | 56.0 |
| TENT [41] | 62.1 ± 4.6 | 49.8 ± 5.0 | 51.6 ± 1.9 | 59.4 ± 1.2 | 53.9 ± 1.0 | 52.2 ± 2.9 | 62.8 ± 3.0 | 56.0 |
| LAME [4] | **33.1 ± 2.4** | **37.8 ± 0.4** | 68.0 ± 8.8 | **37.1 ± 6.7** | 73.2 ± 2.6 | 39.0 ± 7.6 | 66.4 ± 4.0 | 50.7 |
| CoTTA [44] | 61.7 ± 4.2 | 50.0 ± 4.9 | 51.6 ± 1.5 | 59.4 ± 1.1 | 54.4 ± 1.0 | 52.4 ± 2.8 | 62.6 ± 2.9 | 56.0 |
| NOTE | 41.7 ± 5.9 | 40.7 ± 0.8 | 55.5 ± 10.8 | 45.8 ± 4.6 | **45.8 ± 10.4** | **32.9 ± 1.1** | 55.5 ± 10.4 | **45.4** |

## B.3 Ablation study

Table 14: Average classification error (%) and their corresponding standard deviations of varying ablation settings on CIFAR10-C with **temporally correlated test streams**, shown per domain. **Bold** fonts indicate the lowest classification errors. Averaged over three runs.

| Method | Gaussian | Shot | Impulse | Defocus | Glass | Motion | Zoom | Snow | Frost | Fog | Brightness | Contrast | Elastic | Pixelate | JPEG | Avg |
|---|---|---|---|---|---|---|---|---|---|---|---|---|---|---|---|---|
| Source | 74.0 ± 3.3 | 66.8 ± 3.5 | 75.3 ± 4.2 | 43.3 ± 2.7 | 48.0 ± 2.7 | 32.6 ± 1.2 | 35.2 ± 2.6 | 22.0 ± 0.4 | 33.0 ± 2.5 | 25.9 ± 0.9 | 8.5 ± 0.3 | 66.1 ± 1.8 | 23.4 ± 0.7 | 53.6 ± 0.7 | 26.8 ± 0.7 | 42.3 |
| IABN | 44.5 ± 2.7 | 41.3 ± 2.3 | 48.0 ± 1.9 | 16.3 ± 1.0 | 39.9 ± 0.1 | 13.8 ± 0.7 | 16.1 ± 0.7 | 14.9 ± 0.3 | 17.8 ± 0.6 | 16.3 ± 0.6 | 7.6 ± 0.2 | 8.8 ± 0.3 | 22.5 ± 0.3 | 34.0 ± 1.2 | 26.7 ± 0.6 | 24.6 |
| PBRS | 45.2 ± 3.0 | 38.5 ± 4.9 | 46.8 ± 3.3 | 24.5 ± 2.2 | 38.2 ± 2.8 | 19.1 ± 0.9 | 20.0 ± 0.2 | 16.5 ± 0.2 | 19.1 ± 2.4 | 16.5 ± 0.4 | **7.1 ± 0.7** | 34.4 ± 3.0 | 21.5 ± 0.5 | 39.8 ± 4.7 | **25.2 ± 0.4** | 27.5 |
| IABN + RS | **33.7 ± 6.4** | **30.0 ± 6.7** | **37.6 ± 2.9** | 13.6 ± 0.3 | 34.9 ± 1.9 | 12.4 ± 1.2 | **14.5 ± 1.7** | **13.9 ± 1.1** | **15.0 ± 3.1** | **14.0 ± 1.3** | 7.2 ± 0.0 | **7.4 ± 0.7** | 21.1 ± 0.9 | **26.2 ± 4.4** | 25.9 ± 1.1 | **20.5** |
| IABN + PBRS | 34.9 ± 1.6 | 32.3 ± 3.1 | 39.6 ± 2.5 | **13.6 ± 0.5** | 35.8 ± 1.9 | **11.8 ± 0.8** | **14.5 ± 0.5** | 14.1 ± 0.6 | 15.2 ± 1.3 | 14.2 ± 0.6 | 7.7 ± 0.3 | 7.6 ± 0.6 | **20.8 ± 0.7** | 27.7 ± 2.6 | 26.4 ± 0.5 | 21.1 |

Table 15: Average classification error (%) and their corresponding standard deviations of varying ablation settings on CIFAR100-C with **temporally correlated test streams**, shown per domain. **Bold** fonts indicate the lowest classification errors. Averaged over three runs.

| Method | Gaussian | Shot | Impulse | Defocus | Glass | Motion | Zoom | Snow | Frost | Fog | Brightness | Contrast | Elastic | Pixelate | JPEG | Avg |
|---|---|---|---|---|---|---|---|---|---|---|---|---|---|---|---|---|
| Source | 88.1 | 86.8 | 93.7 | 64.9 | 79.7 | 55.5 | 57.7 | 53.8 | 66.3 | 59.3 | 33.0 | 81.4 | 49.2 | 73.6 | 55.5 | 66.6 |
|  | ± 0.2 | ± 0.6 | ± 0.6 | ± 0.4 | ± 0.9 | ± 0.3 | ± 0.2 | ± 0.4 | ± 0.8 | ± 0.4 | ± 0.3 | ± 0.4 | ± 0.4 | ± 1.1 | ± 0.3 |  |
| IABN | 79.3 | 77.2 | 84.2 | 45.0 | 69.6 | 40.9 | 43.1 | 42.5 | 48.6 | 52.5 | 30.4 | 40.5 | 47.6 | 59.8 | 56.2 | 54.5 |
|  | ± 0.7 | ± 0.7 | ± 1.0 | ± 0.6 | ± 0.3 | ± 0.3 | ± 0.6 | ± 0.4 | ± 0.3 | ± 0.5 | ± 0.1 | ± 0.7 | ± 0.5 | ± 1.1 | ± 0.4 |  |
| PBRS | 68.8 | 66.2 | 73.3 | 46.2 | 64.9 | 41.8 | 41.7 | 44.2 | 48.5 | 44.7 | **28.3** | 60.1 | **44.2** | 51.9 | **50.5** | 51.7 |
|  | ± 0.6 | ± 0.4 | ± 0.9 | ± 0.6 | ± 1.5 | ± 0.6 | ± 0.3 | ± 0.4 | ± 0.7 | ± 0.2 | **± 0.2** | ± 0.4 | **± 0.4** | ± 0.8 | **± 0.5** |  |
| IABN + RS | 66.8 | 65.2 | 73.1 | 38.7 | 63.0 | 36.6 | 38.0 | 41.9 | 43.9 | 44.6 | 29.5 | 33.5 | 46.0 | 49.9 | 52.4 | 48.2 |
|  | ± 2.1 | ± 0.3 | ± 1.0 | ± 0.4 | ± 0.9 | ± 0.0 | ± 0.2 | ± 0.8 | ± 0.4 | ± 0.5 | ± 0.3 | ± 0.7 | ± 0.5 | ± 0.9 | ± 0.4 |  |
| IABN + PBRS | **66.2** | **64.2** | **72.6** | **37.2** | **61.1** | **35.4** | **37.4** | **40.0** | **42.5** | **43.4** | 29.4 | **32.1** | 44.3 | **47.5** | 51.3 | **47.0** |
|  | **± 0.8** | **± 1.6** | **± 0.4** | **± 0.8** | **± 0.7** | **± 0.3** | **± 0.4** | **± 0.4** | **± 0.3** | **± 0.5** | ± 0.1 | **± 0.5** | ± 0.4 | **± 0.6** | ± 0.3 |  |

Table 16: Average classification error (%) and their corresponding standard deviations of varying ablation settings on CIFAR10-C with **uniformly distributed test streams**, shown per domain. **Bold** fonts indicate the lowest classification errors. Averaged over three runs.

| Method | Gaussian | Shot | Impulse | Defocus | Glass | Motion | Zoom | Snow | Frost | Fog | Brightness | Contrast | Elastic | Pixelate | JPEG | Avg |
|---|---|---|---|---|---|---|---|---|---|---|---|---|---|---|---|---|
| Source | 74.0 | 66.8 | 75.3 | 43.3 | 48.0 | 32.6 | 35.2 | 22.0 | 33.0 | 25.9 | 8.5 | 66.1 | 23.4 | 53.6 | 26.8 | 42.3 |
|  | ± 3.3 | ± 3.5 | ± 4.2 | ± 2.7 | ± 2.7 | ± 1.2 | ± 2.6 | ± 0.4 | ± 2.5 | ± 0.9 | ± 0.3 | ± 1.8 | ± 0.7 | ± 0.7 | ± 0.7 |  |
| IABN | 44.5 | 41.4 | 48.1 | 16.3 | 39.9 | 13.9 | 16.2 | 14.9 | 17.9 | 16.4 | 7.6 | 8.8 | 22.5 | 34.1 | 26.7 | 24.6 |
|  | ± 2.7 | ± 2.3 | ± 1.9 | ± 1.0 | ± 0.1 | ± 0.7 | ± 0.7 | ± 0.3 | ± 0.6 | ± 0.5 | ± 0.2 | ± 0.3 | ± 0.4 | ± 1.2 | ± 0.6 |  |
| PBRS | 43.4 | 37.9 | 46.2 | 21.8 | 36.8 | 18.1 | 17.6 | 16.1 | 19.3 | 15.2 | **7.1** | 32.5 | **20.0** | 30.7 | **23.8** | 25.8 |
|  | ± 0.8 | ± 0.6 | ± 1.5 | ± 2.0 | ± 1.0 | ± 0.3 | ± 0.8 | ± 0.8 | ± 0.1 | ± 0.5 | **± 0.4** | ± 1.5 | **± 0.2** | ± 0.7 | **± 0.1** |  |
| IABN + RS | 33.8 | 31.1 | 40.4 | 13.3 | 35.6 | 11.8 | 13.2 | 14.6 | **14.9** | 14.7 | 7.7 | 8.1 | 22.3 | **24.6** | 25.1 | 20.7 |
|  | ± 1.6 | ± 0.9 | ± 1.3 | ± 0.7 | ± 0.2 | ± 0.6 | ± 0.3 | ± 0.3 | **± 0.6** | ± 0.4 | ± 0.2 | ± 0.4 | ± 0.5 | **± 1.9** | ± 1.2 |  |
| IABN + PBRS | **33.5** | **30.0** | **38.2** | **12.6** | **34.4** | **11.5** | **12.9** | **14.1** | 15.2 | **14.0** | 7.4 | **7.8** | 20.7 | 24.7 | 24.2 | **20.1** |
|  | **± 1.7** | **± 1.6** | **± 0.9** | **± 0.8** | **± 0.8** | **± 0.5** | **± 0.6** | **± 0.2** | ± 0.8 | **± 0.6** | ± 0.2 | **± 0.2** | ± 0.3 | ± 0.7 | ± 0.4 |  |

Table 17: Average classification error (%) and their corresponding standard deviations of varying ablation settings on CIFAR100-C with **uniformly distributed test streams**, shown per domain. **Bold** fonts indicate the lowest classification errors. Averaged over three runs.

| Method | Gaussian | Shot | Impulse | Defocus | Glass | Motion | Zoom | Snow | Frost | Fog | Brightness | Contrast | Elastic | Pixelate | JPEG | Avg |
|---|---|---|---|---|---|---|---|---|---|---|---|---|---|---|---|---|
| Source | 88.1 | 86.8 | 93.7 | 64.9 | 79.7 | 55.5 | 57.7 | 53.8 | 66.3 | 59.3 | 33.0 | 81.4 | 49.2 | 73.6 | 55.5 | 66.6 |
|  | ± 0.2 | ± 0.6 | ± 0.6 | ± 0.4 | ± 0.9 | ± 0.3 | ± 0.2 | ± 0.4 | ± 0.8 | ± 0.4 | ± 0.3 | ± 0.4 | ± 0.4 | ± 1.1 | ± 0.3 |  |
| IABN | 79.3 | 77.2 | 84.3 | 45.0 | 69.6 | 40.9 | 43.1 | 42.5 | 48.6 | 52.5 | 30.5 | 40.5 | 47.6 | 59.8 | 56.2 | 54.5 |
|  | ± 0.6 | ± 0.6 | ± 1.0 | ± 0.5 | ± 0.2 | ± 0.3 | ± 0.6 | ± 0.4 | ± 0.3 | ± 0.5 | ± 0.1 | ± 0.7 | ± 0.5 | ± 1.1 | ± 0.4 |  |
| PBRS | 68.6 | 66.0 | 72.9 | 45.3 | 64.1 | 40.9 | 41.6 | 43.7 | 47.9 | 44.2 | **28.3** | 59.9 | 44.2 | 51.1 | **50.4** | 51.3 |
|  | ± 1.0 | ± 0.3 | ± 0.3 | ± 0.3 | ± 0.8 | ± 0.5 | ± 0.5 | ± 0.2 | ± 0.2 | ± 0.3 | **± 0.3** | ± 0.7 | ± 0.5 | ± 1.6 | **± 0.6** |  |
| IABN + RS | 67.1 | 65.6 | 74.0 | 39.0 | 61.4 | 36.5 | 38.7 | 41.4 | 44.0 | 45.0 | 30.0 | 34.0 | 46.0 | 48.8 | 52.5 | 48.3 |
|  | ± 1.2 | ± 0.3 | ± 0.4 | ± 0.3 | ± 1.3 | ± 0.1 | ± 0.8 | ± 0.2 | ± 0.4 | ± 0.2 | ± 0.2 | ± 0.2 | ± 1.4 | ± 1.3 | ± 0.5 |  |
| IABN + PBRS | **65.6** | **62.6** | **72.0** | **36.8** | **60.5** | **34.9** | **36.7** | **39.6** | **41.7** | **42.3** | 28.6 | **32.3** | **43.8** | **47.7** | 50.9 | **46.4** |
|  | **± 1.0** | **± 0.7** | **± 0.2** | **± 0.7** | **± 0.7** | **± 0.5** | **± 0.2** | **± 0.2** | **± 0.6** | **± 0.3** | ± 0.2 | **± 0.9** | **± 0.2** | **± 0.4** | ± 0.2 |  |

# C  Replacing BN with IABN during test time

Table 18: Average classification error (%) and corresponding standard deviations of varying ablation settings on CIFAR10-C/100-C under temporally correlated (non-i.i.d.) and uniformly distributed (i.i.d.) test data stream. IABN* refers to replacing BN with IABN during test time (no pre-training with IABN layers). **Bold** fonts indicate the lowest classification errors. Averaged over three runs.

| Method | **Temporally correlated test stream** | | | **Uniformly distributed test stream** | | |
|---|---|---|---|---|---|---|
| | CIFAR10-C | CIFAR100-C | Avg | CIFAR10-C | CIFAR100-C | Avg |
| Source | 42.3 ± 1.1 | 66.6 ± 0.1 | 54.4 | 42.3 ± 1.1 | 66.6 ± 0.1 | 54.4 |
| **IABN*** | 27.1 ± 0.4 | 60.8 ± 0.1 | 44.0 | 27.1 ± 0.4 | 60.8 ± 0.2 | 44.0 |
| IABN | 24.6 ± 0.6 | 54.5 ± 0.1 | 39.5 | 24.6 ± 0.6 | 54.5 ± 0.1 | 39.5 |
| **IABN*+PBRS** | 24.9 ± 0.2 | 55.9 ± 0.2 | 40.4 | 23.2 ± 0.4 | 55.3 ± 0.1 | 39.3 |
| IABN+PBRS | **21.1 ± 0.6** | **47.0 ± 0.1** | **34.0** | **20.1 ± 0.5** | **46.4 ± 0.0** | **33.2** |

For pre-trained models with BN layers such as ResNet [12], NOTE needs to re-train the model by replacing BN layers with IABN layers in order to utilize the effectiveness of IABN. This requires the additional computational cost of re-training, which might make it inconvenient to utilize off-the-shelf models. We further investigate whether simply switching BN to IABN without re-training still leads to performance gain.

Table 18 shows the result of this experiment, where IABN* refers to replacing BN with IABN during test time. We note that IABN* still shows a significant reduction of errors under CIFAR10-C and CIFAR100-C datasets compared with BN (Source). We interpret this as the normalization correction in IABN is somewhat valid without re-training the model. We notice that IABN* outperforms the baselines in CIFAR10-C with 27.1% error, while the second best (LAME) shows 36.2% error [19]. In addition, IABN* also shows improvement combined with PBRS. This implies that IABN can be used without re-training the model, which aligns with the fully test-time adaptation paradigm introduced in a recent study [41].

# D  License of assets

**Datasets**  KITTI dataset (CC-BY-NC-SA 3.0), KITTI-rain dataset (CC-BY-NC-SA 3.0), CIFAR10, 100 (MIT License), ImageNet-C (Apache 2.0), MNIST-C (CC-BY-NC-SA 4.0), HARTH dataset (MIT License), and the Extrasensory dataset (CC-BY-NC-SA 4.0)

**Codes**  Code for rain augmentation on the KITTI dataset (Apache 2.0), torch-vision for ResNet18 and ResNet50 (Apache 2.0), code for depth estimation used in rain augmentation on the KITTI dataset (UCLB ACP-A License), code for generating Dirichlet distributions (Apache 2.0), the official repository of CoTTA (MIT License), the official repository of TENT (MIT License), and the official repository of LAME (CC BY-NC-SA 4.0).

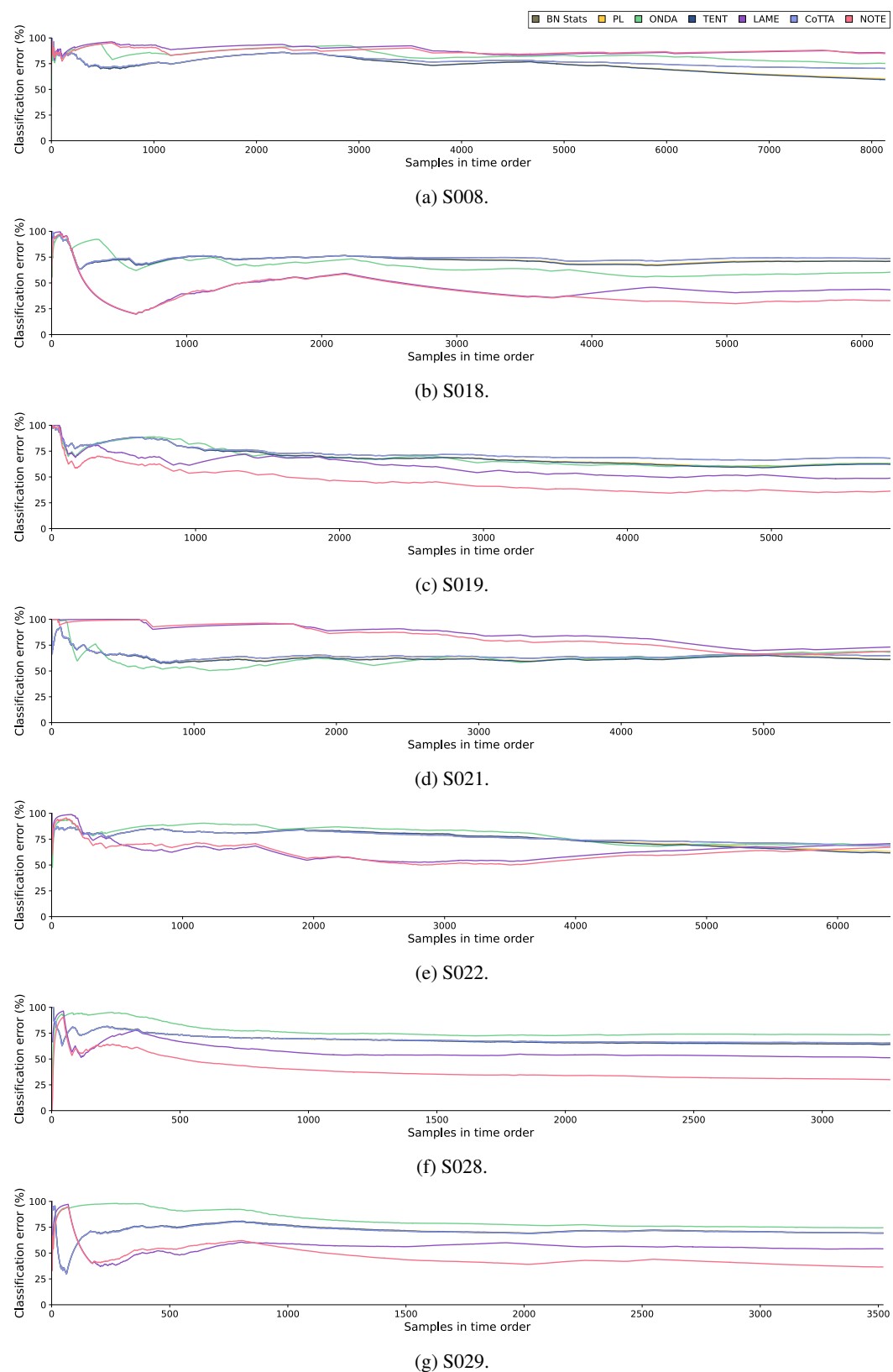

(a) S008.

(b) S018.

(c) S019.

(d) S021.

(e) S022.

(f) S028.

(g) S029.

Figure 7: Illustration of the real-time cumulative classification error change of different methods on the HARTH dataset. The x-axis denotes the samples in order, whereas the y-axis denotes the error rate in percentage. Note that some lines are not clearly visible due to overlap.

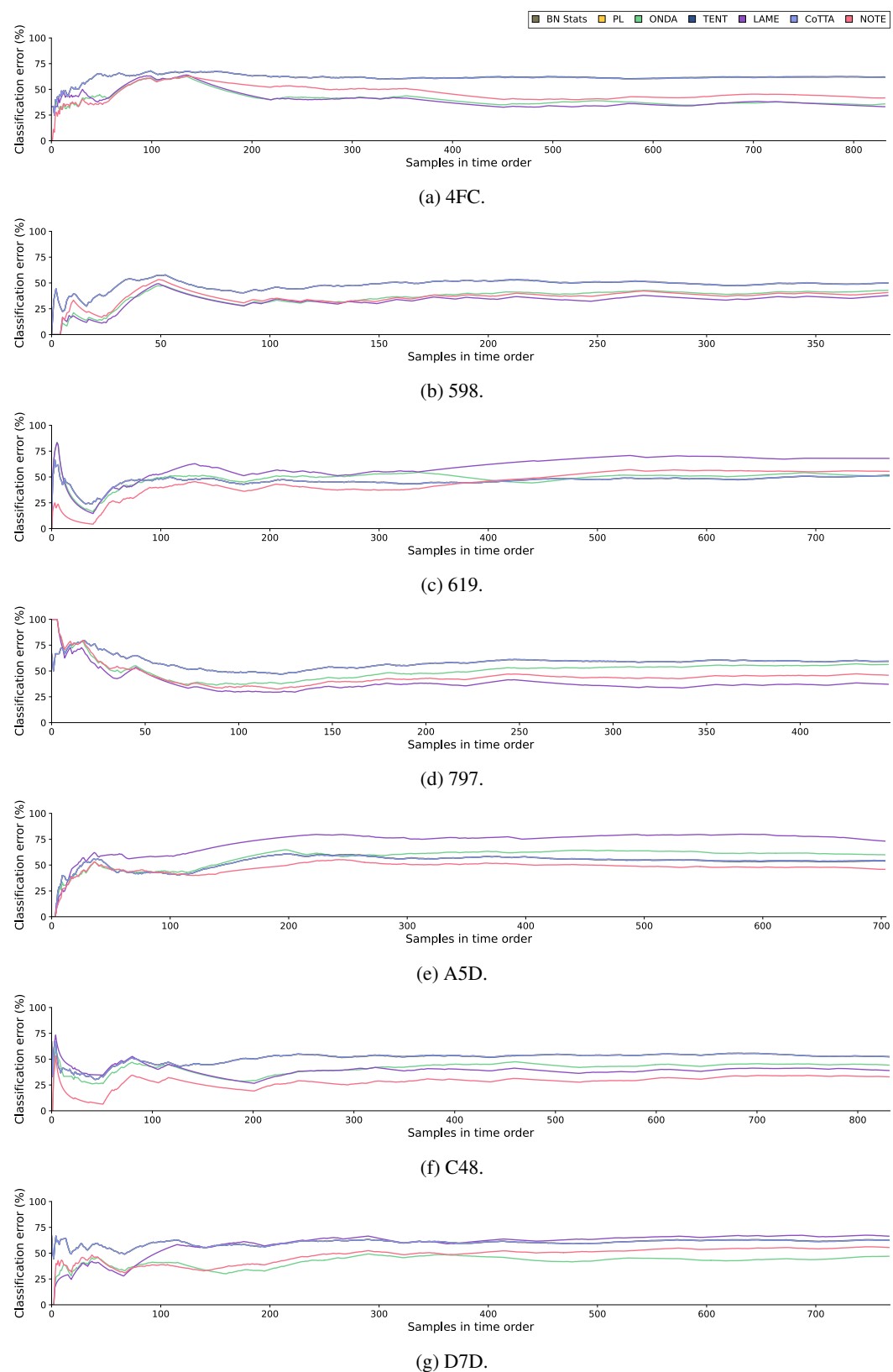

Figure 8: Illustration of the real-time cumulative classification error change of different methods on the Extrasensory dataset. The x-axis denotes the samples in order, whereas the y-axis denotes the error rate in percentage. Note that some lines are not clearly visible due to overlap.