# OpenReview forum: "NOTE: Robust Continual Test-time Adaptation Against Temporal Correlation"
_NeurIPS.cc/2022/Conference — NeurIPS 2022 Accept_

### Official Review · Reviewer_AWKz · 2022-07-11

**Rating:** 6
**Confidence:** 4
**Soundness:** 3 good
**Presentation:** 2 fair
**Contribution:** 3 good

**Summary:**

Test-time training is an emerging and promising approach for building robust machine learning models to distribute shifts. This paper claims that most existing TTA methods assume that the test samples come from i.i.d distribution, but it does not hold for many practical applications of TTA, such as self-driving, human activity recognition, etc. This paper then proposes two new techniques to adapt the model under non-i.i.d setup, namely Instance-Aware Batch Normalization and Prediction-Balanced Reservoir Sampling (PBRS). Experimental results on synthesized non-i.i.d setup and realistic non-i.i.d setup show that the proposed method significantly outperformed the existing methods.


**Questions:**

(1) Why do we need equation (4) and equation (5)? What happens if we always u_{b, c} to compute outputs?

(2) What is the final output of IABN layer?

(3) Regarding the exponential moving average presented in 3.2, do you stop gradient for the past mean and sigma?

(4) Why do we need lines 9-17 for PBRS? In other words, why don’t you randomly discard one instance in memory even if y_t \in L?

(5) In table 1, why does the value for Source differ between i.i.d setup and non-i.i.d setup?

(6) If I correctly understood the paper, PBRS can be incorporated with any prior studies, e.g., TENT, PL. Why did you choose to combine it with IABN? In other words, is IABN itself perform better than existing methods?


**Limitations:**

This paper adequately addressed the limitation and potential negative social impact.

**Strengths And Weaknesses:**

---
**Strengths**

(1) Test-time adaptation is a timely and important topic. This paper pointed out important issues of existing TTA methods.

(2) This paper validates the effectiveness of the proposed method on both synthetic and realistic benchmark tasks of non-i.i.d adaptation setup.

---
**Weaknesses**

Overall, I think this paper provides a good contribution for the test-time adaptation research, with extensive experiments. However, I found several important details lacking in current manuscripts, including the detail of the proposed method (especially IABN part), explanation about why the proposed method should work better than prior works, and empirical validations.  See questions for the detailed comments.

Minor comments/questions:
- Equation (4), the left hand side should be u_c/L?
- Lines 282-284, can you explain why LAME does not works well on i.i.d setup, or even worse than non-i.i.d setup?

---

> ### Author Response · Authors · 2022-08-02
> **Official response to Reviewer AWKz (2/2)**
>
> ---
> **Q6. (6) If I correctly understood the paper, PBRS can be incorporated with any prior studies, e.g., TENT, PL. Why did you choose to combine it with IABN? In other words, is IABN itself perform better than existing methods?**
>
>
> You are right. Theoretically, PBRS can be combined with baselines (and IABN also can be). However, our motivation for the joint use of IABN and PBRS is to solve two different problems in test-time adaptation under temporally correlated test streams. As we wrote in our original submission line 112-120:
> > Under scenarios where test data are temporally correlated, however, naïvely adapting to the incoming batch of test samples [28 , 32, 40, 43] could be problematic for both approaches: the batch is now more likely to (a) remove **instance-wise variations** that are actually useful to predict $y$, i.e., the “contents” rather than “styles” through normalization, and (b) include a **bias in $p(y)$ rather than uniform**, which can negatively affect the test-time adaptation objective such as entropy minimization. We **propose two approaches to tackle each of the failure modes of adapting BN under temporal correlation**. Our method consists of two components: (a) Instance-Aware Batch Normalization (IABN) (Section §3.1) to overcome the limitation of BN under distribution shift and (b) Prediction-Balanced Reservoir Sampling (PBRS) (Section §3.2) to combat with the temporal correlation of test batches.
>
>
> As demonstrated in our ablation study (Table 3), IABN-only still shows the lowest error in CIFAR10-C (24.6%) and the second lowest error in CIFAR100-C (54.5%), under temporally correlated test streams. When IABN and PBRS are jointly used, the error becomes even lower for both CIFAR10-C (21.1%) and CIFAR100-C (47.0%), which highlight the synergy between them against temporally correlated streams.
>
>
>
>
> ---
> **Q7 (minor). Equation (4), the left hand side should be u_c/L?**
>
>
> Our Equation (4) is correct - $s^2\_{\tilde{\mu},c} := \frac{\bar{\sigma}^2\_c}{L}$. The sample mean, $\tilde{\mu}\_{b,c}$,  follows the *sampling distribution* of sample size $L$ in $\mathcal{N}(\bar{\mu}, \bar{\sigma}^2)$ as the population, as in line 136 of our original submission. Thus, the “variance" of the sample mean, $s^2_{\tilde{\mu},c}$, is driven from the variance $\bar{\sigma}^2$ of the distribution where it is sampled from.
>
>
>
>
>
>
> ---
> **Q8 (minor). Lines 282-284, can you explain why LAME does not works well on i.i.d setup, or even worse than non-i.i.d setup?**
>
>
> LAME is a method that directly manipulates the output feature vector with laplacian optimization to achieve good performance in non-i.i.d. settings. In essence, as stated in its paper [4], what it does is that it “discourages deviations from the predictions of the pre-trained model”. Thus, it thrives in non-i.i.d scenarios where consecutive samples are correlated, which can cause the model to “overspecialize” in a narrow portion of the target stream. However, it is not helpful in i.i.d. scenarios where such an assumption does not hold, causing - quoting its paper: **“it does not noticeably help in i.i.d and class-balanced scenarios”**.
>
> We added the explanation of why LAME does not work in the i.i.d. scenarios in the revised draft (Section 4.1).

---

> ### Author Response · Authors · 2022-08-02
> **Official response to Reviewer AWKz (1/2)**
>
> We sincerely appreciate your effort and time in offering us thoughtful comments. We respond to each of your questions and concerns one-by-one in what follows. We also ask you to kindly refer to the *common response* we have posted together.
>
>
> ---
> **Comment1. Overall, I think this paper provides a good contribution for the test-time adaptation research, with extensive experiments. However, I found several important details lacking in current manuscripts, including the detail of the proposed method (especially IABN part), explanation about why the proposed method should work better than prior works, and empirical validations. See questions for the detailed comments.**
>
> We thank Reviewer AWKz for all your clarifying questions that help us to improve the readability of our method section. Please refer to our answers below. We also submitted a revised draft that addressed them.
>
>
> ---
> **Q1. (1) Why do we need equation (4) and equation (5)? What happens if we always u_{b, c} to compute outputs?**
>
> Equation (4) and Equation (5) are the key ideas of IABN that make it robust to distributional changes during test time. It corrects the learned batch normalization statistics when the current input is too deviated from the learned distribution.
> If we use the sample mean $\tilde{\mu}\_{b,c}$ and the sample variance $\tilde{\sigma}^2\_{b,c}$ instead, it is the same as using Instance Normalization, as we originally put in line 143-144:
> > If one chooses too small $k\geq 0$, IABN may remove useful features, e.g., styles, of input (as with IN), which can degrade the overall classification (or regression) performance [24].
>
>
> ---
> **Q2. (2) What is the final output of IABN layer?**
>
> The output of the IABN layer is similar to BN, except for the notations used in Equation (5). Specifically, IABN replaces  $\mu\_c$ and $\sigma\_c^2$ in BN with ${\mu}\_{b,c}^{IABN}$ and $({\sigma}\_{b,c}^{ IABN})^2$, respectively:
>
>
> - $\mathrm{BN}(\mathbf{f}\_{:,c,:}; \mu\_c, \sigma^2\_c) := \gamma\cdot\frac{\mathbf{f}\_{:,c,:} - \mu\_c}{\sqrt{\sigma\_c^2 + \epsilon}} + \beta$  (same as Equation (1))
>
> - $\mathrm{IABN}(\mathbf{f}\_{b,c,:}; \bar{\mu}\_c, \bar{\sigma}^2\_c; \tilde{\mu}\_{b,c}, \tilde{\sigma}^2_{b,c}) := \gamma\cdot\frac{\mathbf{f}\_{b,c,:} - {\mu}\_{b,c}^{\tt IABN}}{\sqrt{({\sigma}\_{b,c}^{\tt IABN})^2 + \epsilon}} + \beta$
>
> We believe describing the final output of the IABN layer would improve the readability of the paper. We included the description of the output of IABN in both Section 3.1 and Figure 3, in the revised draft.
>
> ---
> **Q3. (3) Regarding the exponential moving average presented in 3.2, do you stop gradient for the past mean and sigma?**
>
> We don’t need to stop; $\mu$ and $\sigma^2$ are not trainable parameters but internal statistics of BN (and IABN) layers. The trainable parameters are $\gamma$ and $\beta$ as we described in the original submission (line 166-169):
> > Specifically, we update the normalization statistics, namely the means $\mu$ and variances $\sigma^2$, via exponential moving average …. We further optimize the affine parameters, scaling factor $\gamma$ and bias term $\beta$ via entropy minimization, similar to a previous study [40].
>
>
> ---
> **Q4. (4) Why do we need lines 9-17 for PBRS? In other words, why don’t you randomly discard one instance in memory even if y_t \in L?**
>
> If we randomly discard one instance in the set of the majority class(es) $L$, old samples hardly survive as they are candidates to be dropped for a long time, and the memory is likely to have relatively new samples. With Reservoir Sampling (line 9-17), we can ensure the samples that reside in the memory are *time-uniform*. This is beneficial for temporally-correlated streams; time-uniform samples are more balanced in terms of distribution than adjacent samples, and thus estimating the target distribution from time-uniform samples helps the generalization of the model.
>
>
> ---
> **Q5. (5) In table 1, why does the value for Source differ between i.i.d setup and non-i.i.d setup?**
>
> Thank you for pointing out this issue. We used an old result for the Avg column in non-i.i.d. Setup. We updated this value in the revised paper.

---

> > ### Comment · Reviewer_AWKz · 2022-08-08
> > **Thank you for your clarification**
> >
> > Thank you for your detailed response. It resolve most concerns, however, I am still not convinced about the necessarily of reservoir sampling. If inputs are temporally correlates, isn't it natural to prioritize temporally local inputs during adaptation? Besides, in most experimental setup, it seems the labels are also temporally correlates, which means balancing class label would eliminates (most) temporal correlations.
> >
> > Overall, I still think the paper provide several interesting findings. I therefore would like to keep my score.

---

> > > ### Author Response · Authors · 2022-08-08
> > > **Thank you for your response**
> > >
> > > Thank you for your response to our rebuttal! We are glad that our rebuttal addressed most of your concerns.
> > >
> > > ---
> > > **ReQ1. I am still not convinced about the necessarily of reservoir sampling. If inputs are temporally correlates, isn't it natural to prioritize temporally local inputs during adaptation?**
> > >
> > > Thank you for your follow-up question. As shown in the results of the baselines in our experiments (Table 1-2), prioritizing temporally local inputs during adaptation” **mostly fails** in temporally correlated scenarios. In the background section (Section 2.2) of our original submission, we mentioned that existing studies use only an incoming batch of samples for adaptation, which is the extreme form of prioritizing temporal samples (line 98-101):
> > > > Although a conventional way of computing BN in test-time is to set $\mu$ and $\sigma^2$ as those estimated from _training_ (or source) data, say $\bar{\mu}$ and $\bar{\sigma}^2$, the state-of-the-art TTA methods based on adapting BN layers [28, 32, 40, 43] instead **use the statistics computed directly from the recent test batch to de-bias distributional shifts** at test-time, ...
> > >
> > > We remark that “prioritizing temporally local inputs” can lead to undesirable (and somewhat unexpected) bias and catastrophic forgetting in our temporally correlated scenarios as explained in our original submission (Section 3, line 112-116):
> > > > Under scenarios where test data are temporally correlated, however, naively adapting to the incoming batch of test samples [28, 32, 40, 43] could be problematic for both approaches: the batch is now more likely to (a) **remove instance-wise variations** that are actually useful to predict $y$, i.e., the “contents” rather than “styles”' through normalization, and (b) **include a bias in $p(y)$ rather than uniform**, which can negatively affect the test-time adaptation objective such as entropy minimization.
> > >
> > > Thus, we rather **aim to eliminate temporal correlation** for better calibration of BN layers, which is achievable via our Prediction-Balanced Reservoir Sampling and results in outperforming the baselines in temporally correlated scenarios.
> > >
> > > ---
> > > **ReQ2. Besides, in most experimental setup, it seems the labels are also temporally correlates, which means balancing class label would eliminates (most) temporal correlations.**
> > >
> > > In addition to our original response in Q4, if temporal correlation is introduced *within a class*, we might lose instance-wise variations, which leads to catastrophic forgetting. For instance, consecutive frames for the “bicycle” class (Figure 8 in the supplementary file) are very similar and thus do not give useful information for class-specific variations.
> > >
> > > ---
> > >
> > > Thank you again for the valuable suggestions and comments.
> > > If you have any remaining suggestions or concerns, please let us know!
> > >
> > > Best,
> > >
> > > Authors.

---

### Official Review · Reviewer_bg8w · 2022-07-11

**Rating:** 5
**Confidence:** 4
**Soundness:** 3 good
**Presentation:** 3 good
**Contribution:** 3 good

**Summary:**

Temporal correlated data violates the i.i.d. assumption and the existing test-time adaptation methods are prone to overfitting under high correlated test data. This work proposed two components, IABN and PBRS, to tackle the temporal correlation. IABN only updates BN parameters when test data distribution is sufficiently different and PBRS resamples data samples for calculating test data batch statistics. Experiments are carried out on simulated temporal correlated data and real datasets.

**Questions:**

Evaluation and discussion on how to generalize to transformer based backbones would further improve the quality of this work.

For streaming test, it would be interesting to report the real-time accuracy which illustrates the real adaptation power in test-on-stream manner.

**Limitations:**

Yes.

**Strengths And Weaknesses:**

Strength:

1. Temporal correlation is a practical concern in test-time training. Addressing the challenges in TTA for temporal correlated data is important.

2. Experiments are extensive, covering both visual and audio modalities.

Weakness:

3. The improvement on KITTI is very marginal (as shown in Tab. 2). KITTI is the real temporally correlated dataset that motivates this work but the improvement is not significant to support the claims.

4. For temporally correlated test-time adaptation task, CIFAR10-C/100-C can hardly simulate the temporal correlation. How does the temporally correlated streams look like? It is important to give illustrations or analysis of why this synthesized data stream presents temporal correlation.

5. Adjusting Batchnorm has limitation to certain backbone networks. For example, ViT (transformer) does not have batchnorm layers, how does this approach apply to transformer backbones where BN is missing?

---

> ### Author Response · Authors · 2022-08-02
> **Official response to Reviewer bg8w (2/2)**
>
> ---
> **Q1. Evaluation and discussion on how to generalize to transformer based backbones would further improve the quality of this work.**
>
> Thank you for your suggestion. We discuss the applicability of our method to Transformer-based architecture in NLP tasks following Reviewer GmfJ who also asked a similar question to yours.
>
> We sincerely appreciate your time and effort in providing us with positive comments. We respond to your question in what follows. We also ask you to kindly refer to the *common response* we have posted together.
>
> ---
> **Comment1. In the experiments, the authors should use more tasks such as some NLP data and more popular base models like Transformer-based ones for both CV and NLP. Otherwise, it's hard to measure how the proposed methods really work in modern AI systems.**
>
> We would like to highlight that all the SOTA baselines (BN Stats, ONDA, PL, TENT, CoTTA) except for LAME, are not applicable to Transformer-based models. Similarly, NOTE is not applicable to models without BN layers. In our original submission, we clarified and discussed this limitation in Section 6:
> > Similar to existing TTA algorithms [28, 32, 40, 43], **we assume that the backbone networks are equipped with BN layers**, and particularly, we replaced the BN layers with IABN layers. While BN is a widely-used component in deep learning, there exist architectures that do not embed BN layers such as LSTMs [13] and Transformers [38]. Whether naively inserting BN or IABN would be sufficient for applying these TTA methods is still in question. A recent study evidenced that BN is advantageous in Vision Transformers [44], showing potential room to apply our idea to architectures without BN layers. However, more in-depth studies are necessary to identify the actual applicability of BN (or IABN) to those architectures.
>
> Nevertheless, the other technical component of our method, PBRS, can be implemented on transformers. To investigate the impact of PBRS on Transformer-based models for NLP tasks, we adopted the **BERT** [r1] tiny model. Specifically, PBRS was utilized to update the trainable parameters in the Layer Normalization layers of the model. We conducted test-time adaptation on common text sentiment analysis datasets, following previous work [r2]; we used  **SST-2** [r3] (movie reviews; source domain) to fine-tune the BERT model and evaluated its test-time adaptation capabilities on **FineFood** [r4] (food reviews; target domain). This setup is valuable for understanding how TTA algorithms perform under the domain gap in text sentiment caused by the difference in topics. During test-time adaptation, we used the Dirichlet distribution for simulating the non-i.i.d. streams, as previously described in our experiment section. The result is presented in the table below:
>
> Table.  Classification error (%) on FineFood with both temporally correlated and uniform test streams. The lower, the better. N/A refers to “not applicable.”
> | | non-i.i.d. | i.i.d. |
> |---|---:|---:|
> | Source | 37.4 ± 0.0 | 37.4 ± 0.0 |
> | BN Stats | N/A | N/A |
> | ONDA | N/A | N/A |
> | PL | N/A | N/A |
> | TENT | N/A | N/A |
> | LAME | 35.7 ± 0.3 | 41.6 ± 0.3 |
> | CoTTA | N/A | N/A |
> | **NOTE (PBRS-only)** | **34.8 ± 2.1** | **34.6 ± 2.5** |
>
> We found that, to some extent, PBRS exhibits its effectiveness. Nevertheless, NOTE cannot fully utilize the synergy between IABN and PBRS regarding balanced statistics updates in this case. While LAME is applicable to models without BN, **LAME’s critical limitation is the performance drop in i.i.d. scenarios**, as shown not only for this particular experiment but also in our main evaluation. The primary reason is, as stated by the authors of LAME, that it “discourages deviations from the predictions of the pre-trained model,'' and thus it “does not noticeably help in i.i.d and class-balanced scenarios.”
>
> In summary, while NOTE shows its effectiveness in **both** non-i.i.d and i.i.d. scenarios, a remaining challenge is to design an algorithm that generalizes to any architecture. We believe the findings and contributions of our work could give valuable insights to future endeavors on this end. We submitted a revised draft to incorporate this discussion.
>
>
> [r1] Devlin, Jacob, et al. "BERT: Pre-training of deep bidirectional transformers for language understanding." arXiv preprint arXiv:1810.04805 2018.
>
> [r2] Moon, Seung Jun, et al. "Masker: Masked keyword regularization for reliable text classification." AAAI 2021.
>
> [r3] Socher, Richard, et al. "Recursive deep models for semantic compositionality over a sentiment treebank." EMNLP 2013.
>
> [r4] McAuley, et al. "From amateurs to connoisseurs: modeling the evolution of user expertise through online reviews." WWW 2013.

---

> > ### Comment · Reviewer_bg8w · 2022-08-08
> > **Thanks for Additional Analysis and Evaluations**
> >
> > Dear Authors,
> >
> > I would like to thank the authors for providing additional analysis and evaluations. After reading other reviewers' comments and the responses and revisions I have two concerns not fully addressed which could be further improved in the future. Firstly, using CIFAR10/100-C to demonstrate test-time adaptation on temporally correlated data as the main results and ablation study do not reflect the full potential of the proposed method, therefore future evaluation should focus on some real datasets which have the temporal correlated feature. Secondly, the proposed IABN is only applicable to architectures with BN layers, which is another restriction to the adoption of IABN in more recent backbone networks. In terms of the hypothesised usercase, i.e. autonomous driving, I also suspect whether TTA is really applicable. Given the high demand in safety, adapting pretrained model on-the-fly is not very likely.
> >
> > Overall, I think this paper has some merits in the problem formulation and I will keep my rating, leaning towards acceptance.

---

> > > ### Author Response · Authors · 2022-08-09
> > > **Thank you for your response**
> > >
> > > Thank you for your response to our rebuttal! We agree that the concerns you pointed out are the current limitations of our study and could be further improved in the future. Still, we believe our contributions make a meaningful step towards the practical applications of the test-time adaptation paradigm, as you acknowledged.
> > >
> > > Thank you again for the valuable suggestions and comments.
> > >
> > > Best,
> > >
> > > Authors.

---

> ### Author Response · Authors · 2022-08-02
> **Official response to Reviewer bg8w (1/2)**
>
> We sincerely appreciate your effort and time in offering us thoughtful comments. We respond to each of your questions and concerns one-by-one in what follows. We also ask you to kindly refer to the *common response* we have posted together.
>
>
> ---
> **Comment1. The improvement on KITTI is very marginal (as shown in Tab. 2). KITTI is the real temporally correlated dataset that motivates this work but the improvement is not significant to support the claims.**
>
> Considering the *relative* improvement of the error with respect to the Source error, NOTE shows around **11%** improvement in KITTI (12.3%→10.9%), **18%** in HARTH (62.6%→51.0%), and **9%** in ExtraSensory (50.2%→45.4%). Thus, we believe that the improvement in KITTI is not marginal. In addition, we emphasize that most baselines (BN Stats, ONDA, PL, TENT, CoTTA)  show even worse error rates after the adaptation.
>
> Furthermore, NOTE consistently shows significant improvements on other datasets (CIFAR10-C: 42.3% →21.1%; CIFAR100-C: 66.6%→ 47.0%; MNIST-C: 16.1%→7.1%, HARTH: 62.6%→51.0% , ExtraSensory: 50.2%→45.4%) while outperforming the baselines. During the rebuttal period, we conducted an additional evaluation on a large-scale robustness benchmark, ImageNet-C, and showed significant improvements (86.1%→80.6%). Please refer to our response to Reviewer 5mFV Q1 for the details of the ImageNet-C result.
>
>
> ---
> **Comment2. For temporally correlated test-time adaptation task, CIFAR10-C/100-C can hardly simulate the temporal correlation. How does the temporally correlated streams look like? It is important to give illustrations or analysis of why this synthesized data stream presents temporal correlation.**
>
> We agree that CIFAR10-C/100-C cannot simulate the genuine temporal correlation. That said, we note that most existing test-time adaptation studies adopt CIFAR10-C/100-C for their main evaluation benchmarks. To acknowledge those previous studies and present a fair comparison with them, we evaluated them under both temporally correlated and uniformly distributed scenarios for CIFAR10-C/100-C. We believe one prominent way to generate temporally correlated distributions from those datasets is to inject class-wise temporal correlation, which can be observed in the class distributions of the real datasets in Figures 1, 8, 9, and 10 in our original submission.

---

### Official Review · Reviewer_5mFV · 2022-07-17

**Rating:** 6
**Confidence:** 5
**Soundness:** 3 good
**Presentation:** 3 good
**Contribution:** 2 fair

**Summary:**

Test-time adaptation updates the model on the test data to improve generalization to shifted data.
While this sort of adaptation can help, most existing methods assume the test data arrives i.i.d., and in particular without temporal dependence.
In temporal settings, such as robotics or autonomous driving, this work first demonstrates the weakness of existing methods, and then proposes a fix (NOn-i.i.d TEst-time adaptation, or NOTE) that adjusts normalization statistics and maintains a reservoir/queue of test data to stabilize test-time optimization.

This fix is an extension of TENT, a test-time adaptation method that updates batch normalization layers in two steps: first, it re-estimates the normalization statistics from test batches, and second, it updates affine scale and shift parameters to minimize the entropy of model predictions.
The normalization part of the fix generalizes instance norm and batch norm (IABN), reducing to each as a special case of the hyperparameters, and in essence correct outliers w.r.t. the batch statistics by using the instance statistics instead (Sec. 3.1).
The reservoir part of the fix adopts reservoir sampling to maintain a queue that is expected to be class-balanced and time-balanced in order to simulate access to i.i.d. data in the non-i.i.d. setting.
The balancing is accomplished by making use of the model predictions as pseudo-labels, in place of the true labels used for continual learning, for prediction based reservoir sampling (PBRS).
With the reservoir samples, current test inputs can be batched with past test inputs, which stabilizes both the normalization statistics and entropy gradients for updating the affine parameters.

Experiments on standard benchmarks for dataset shift (CIFAR-10/100-C), real temporal data with synthetic shift (KITTI), and lesser-known temporal datasets (HARTH, ExtraSensory) demonstrate that existing methods degrade in the temporal setting.
Matched comparison on standard and temporal/class-ordered CIFAR-10/100-C confirm that NOTE achieves comparable accuracy in both settings, rivaling existing methods in the i.i.d. case and signifcantly improving (~50% relative error) in the temporal case.
On the real temporal data (KITTI, HARTH, ExtraSensory) NOTE still does as well as better, although the KITTI result is within the error bars, while HARTH and ExtraSensory are not well-established as benchmarks for shift.
The main components of the method, IABN and PBRS, are ablated and the sensitivity of methods to temporal dependence w.r.t. the degree of non-uniformity and batch size are analyzed.

NOTE is the first work to identify the failure of test-time adaptation on temporal data and achieves state-of-the-art accuracies compared to existing methods in this setting.

**Questions:**

Questions

- How does NOTE perform on ImageNet-C in the i.i.d. and non-i.i.d. settings with ResNet-18 (and ideally ResNet-50)? For the non-i.i.d. setting, consider the same proposed dirichlet ordering, or simply ordering the data by classes.
- What is the full impact of PBRS on the amount of computation for prediction and adaptation? In particular, how much more time is needed per test input? Given that it is used for batching inputs, it seems that forward and backward would require linearly more time in the size of the reservoir. To put it another way, does using a reservoir of size 64 for a test input mean that inference is 64x times slower or at least requires that many more forward passes?
- Is it possible to adopt IABN during testing without altering training? That is, can NOTE operate in a fully test-time manner without having to (re-)train the model with IABN?
- For the real-world datasets, what is the source accuracy on the source data? This is important to report for measuring adaptation so that the severity of the shift and how much it is mitigated can be gauged. As presented, it is hard to know if HARTH and ExtraSensory are hard problems, and if the source model is not already doing quite well.
- (Minor) How are predictions made on the first samples before the reservoir is filled? Is adaptation deferred until the reservoir is full?

Other Feedback

- Are the KITTI, HARTH, and ExtraSensory datasets "real distributions with domain shift" (as in Sec. 4.2)? KITTI-Rain is real data, with a synthetic shift. HARTH and ExtraSensory are both real data, but the shift is not evident.
- Please clarify the hyperparameters for IABN in Equation 5 and the definition of $\psi$ that follows it. $k$ is often used for integer values, but in this case it seems to be a real-valued weight. Consider other notation, such as $alpha$, and introducing this weighting in its own sentence following Equation 5.
- Figure 3 tries to cover a lot, and is less accessible as a result. Part (3) on PBRS does not communicate a lot, as it just shows updated statistics, so consider dropping it in favor of focusing more on the IABN update. Consider more labeling of part (1) to express how IABN corrects BN or not.
- In Figure 3 (2), is the time arrow reversed? I would expect the latest sample $x_t$ to be new while the discarded sample $x_i$ should be old.
- To report failures in the temporal setting, consider highlighting results that are worse than the source model with special formatting. For example, these could be underlined, or typset in red.

**Limitations:**

The limitations covered are thoughtful, in particular when it comes to the potential to amplify bias in testing data. The reliance on batch normalization layers, at least as empirically examined, is honestly discussed. While this discussion is adequate, it would be further improved by underlining the need to alter training, in substituting BN with IABN, as other methods are agnostic to training (like TENT and BN).

**Strengths And Weaknesses:**

Strengths

- Well-motivated: Temporal data is ubiquitous, and many cases that may need adaptation are indeed temporal, such as autonomous driving in anomalous weather or robotics applications in poorly-controlled conditions.
  The difficulty current methods have with temporal data is neatly summarized by Fig. 2, which shows the stark contrast between the i.i.d. and non-i.i.d. settings on a simple dataset (CIFAR-10-C).
- Clear illustrations: Fig. 1 shows two varieties of temporal data, with inputs and labels, where both clearly show the type of temporal correlation higlighted by this work.
- Sound experimental design on standard datasets: The use of CIFAR-10/100-C is standard and serves as a sanity check of the method in the established setting. The application of the Dirichlet distribution for sampling more dependent and less dependent orderings of the data according to its concentration hyperparameter is appropriate and effective for sweeping across degrees of dependence.
- Strong enough results: NOTE rivals existing methods in the existing setting, and does far better in the temporal setting it focuses on.
- Baselines: The experimental comparisons include the basic method, TENT, which is extended by NOTE, alongside the recent and strongest methods such as LAME and CoTTA which were only recently published at CVPR'22.
- Consistency: The same hyperparameters, such as the threshold for IABN, are used across experiments

Weaknesses

- NOTE has to alter training by replacing BN with IABN, and therefore cannot be applied to off-the-shelf models without re-training.
  The test-time adaptation setting is in part motivated by not having access to the source/training data, so this makes NOTE less of a good fit for this particular assumption.
- There is no large-scale evaluation in terms of dataset or model. For data, one would expect results on ImageNet-C, which serves as a gold standard benchmark of robustness to natural shifts like corruptions. For models, ResNet-50 is a common choice among test-time adaptation methods, including those compared to in this work.
- How NOTE updates is not fully specified. While Sec. 3.2 explains that inputs are sampled from the reservoir and batched with test inputs, it does not detail the proportion of test inputs to reservoir inputs and the number of gradient steps among other such considerations.
- The choice of datasets is not entirely justified. HARTH and ExtraSensory are indeed temporal, but at the same time they are unfamiliar for the purpose of evaluating adaptation. Are these datasets subject to shift across the given sources and targets? It would be reassuring to measure source accuracy on source and target to evidence drops and thus the presence of shift.
  The architectures for these datasets are likewise not standard, which would be fine, except the text does not explain how to they incorporate IANB (though this presumably follows the convolution layers).

---

> ### Author Response · Authors · 2022-08-02
> **Official response to Reviewer 5mFV (4/4)**
>
> ---
> **Q5. How are predictions made on the first samples before the reservoir is filled? Is adaptation deferred until the reservoir is full?**
>
> NOTE  **predicts each sample regardless of the memory occupancy**. As discussed in Q2, NOTE does not need to wait for a batch of samples for inference. The samples in memory are utilized only to adapt IABN parameters with balanced samples. We clarified this in the revised draft (Section 3.3).
>
> ---
> **Q6. Are the KITTI, HARTH, and ExtraSensory datasets "real distributions with domain shift" (as in Sec. 4.2)? KITTI-Rain is real data, with a synthetic shift. HARTH and ExtraSensory are both real data, but the shift is not evident.**
>
> We agree that it might be misleading. In the revised draft and supplementary file (Section 4.2 and C.2.2), we replaced “real distributions with domain shift” with “real-distributions with domain shift”. We intended to emphasize that the distributions are real, while domain shift is synthetic for KITTI-Rain and real for HARTH and ExtraSensory. Regarding the evidence of the shift, please refer to Q4.
>
> ---
> **Q7. Please clarify the hyperparameters for IABN in Equation 5 and the definition of ψ that follows it. k is often used for integer values, but in this case it seems to be a real-valued weight. Consider other notation, such as alpha, and introducing this weighting in its own sentence following Equation 5.**
>
> In our original submission, we explained the hyperparameter k and the definition of $\psi$ right after Equation (5), in line 141~: “where ψ (x;λ) =...... k >= 0 is a hyperparameter….”.
>
> We revised the draft to further clarify this (Section 3.1). In addition, we replaced $k$ with $\alpha$ following your suggestion.
>
>
> ---
> **Q8. Figure 3 tries to cover a lot, and is less accessible as a result. Part (3) on PBRS does not communicate a lot, as it just shows updated statistics, so consider dropping it in favor of focusing more on the IABN update. Consider more labeling of part (1) to express how IABN corrects BN or not.**
>
> Thank you for your comment. We described the procedure of IABN better in Figure 3 and in the revised draft.
>
> ---
> **Q9. In Figure 3 (2), is the time arrow reversed? I would expect the latest sample xt to be new while the discarded sample xi should be old.**
>
> Thank you for pointing it out. “Old” and “New” labels in Figure 3 (2) only apply to the samples within the memory. We agree that it might be confusing. We fixed the issue in Figure 3 in the revised draft.
>
> ---
> **Q10. To report failures in the temporal setting, consider highlighting results that are worse than the source model with special formatting. For example, these could be underlined, or typset in red.**
>
> We appreciate your suggestion. We updated the values in tables with red fonts in the revised draft and supplementary file.

---

> > ### Comment · Reviewer_5mFV · 2022-08-08
> > **Thank you for the thorough response!**
> >
> > Please note that this message acknowledges all four of the responses to the initial review, and I am simply making one post to avoid triggering multiple notifications.
> >
> > **Summary**: I have raised my score by one as the response has clarified the points of confusion and has provided additional results that (1) compare on the benchmark standard of ImageNet-C and (2) justify the choice of datasets and claims about shift. Most importantly, the responses to Q1, Q2, and Q4 convincingly resolve the potential issues with not benchmarking at larger scale (Q1), possible increase in computational cost (Q2), and whether or not the user activity datasets exhibited shift (Q4).
> >
> > Having highlighted these improvements, I should note that revision will be needed if this submission is accepted to prioritize results between the main paper and supplement and to ensure that there is space for related work. I encourage the authors to include their results on ImageNet-C, and to further scale them up to ResNet-50, as that is the most commonly reported combination of dataset and model for test-time adaptation.

---

> > > ### Author Response · Authors · 2022-08-09
> > > **Thank you for your response!**
> > >
> > > Thank you for carefully reviewing our rebuttal and leaving an acknowledging response to us. We are glad that our rebuttal addressed your concerns.
> > >
> > > In our final draft, we will incorporate your suggestions. Thank you again for your valuable comments and feedback to improve our work.
> > >
> > > Best,
> > >
> > > Authors.

---

> ### Author Response · Authors · 2022-08-02
> **Official response to Reviewer 5mFV (3/4)**
>
> ---
> **Q2. What is the full impact of PBRS on the amount of computation for prediction and adaptation? In particular, how much more time is needed per test input? Given that it is used for batching inputs, it seems that forward and backward would require linearly more time in the size of the reservoir. To put it another way, does using a reservoir of size 64 for a test input mean that inference is 64x times slower or at least requires that many more forward passes?**
>
> PBRS (and NOTE accordingly) does not incur additional forward passes during inference; it requires **only a single forward-pass** for each test sample. In addition, the number of forward/backward passes is fixed (= number of test samples) and **does not linearly increase according to the size of the memory**.
>
> The additional computational overhead caused by PBRS is two-fold: (1) For every sample, it decides whether to add it to memory via Algorithm 1 without any additional forward/backward passes.  (2) For every 64 samples(= memory size), PBRS updates only the IABN’s normalization statistics ($\bar{\mu}$ and $\bar{\sigma}^2$) via exponential moving average and affine parameters ($\gamma$ and $\beta$) via backward passes; thus the number of samples seen during the test time is equal to the number of samples subject to backward passes.
>
> We emphasize NOTE is computationally efficient compared with state-of-the-art approaches that require multiple forward passes to infer a single sample, such as CoTTA.
>
> We submitted a revised draft to reflect our answer for your comment to highlight the computational advantages of NOTE compared with the baselines.
>
> ---
> **Q3. Is it possible to adopt IABN during testing without altering training? That is, can NOTE operate in a fully test-time manner without having to (re-)train the model with IABN?**
>
> We investigated whether switching BN to IABN without re-training still leads to performance gain. IABN* refers to replacing BN with IABN during test time.
>
> CIFAR dataset in non-i.i.d settings:
> | Method | CIFAR10-C | CIFAR100-C | _Avg_ |
> |---|:---:|:---:|:---:|
> | Source | 42.3 ± 1.1 | 66.6 ± 0.1 | 54.43 |
> | **IABN*** | 27.1 ± 0.4 | 60.8 ± 0.1 | 43.98 |
> | IABN | 24.6 ± 0.6 | 54.5 ± 0.1 | 39.54 |
> | __IABN*+PBRS__ | 24.9 ± 0.2 | 55.9 ± 0.2 | 40.41 |
> | IABN+PBRS | **21.1 ± 0.6** | **47.0 ± 0.1** | **34.03** |
>
> CIFAR dataset in i.i.d settings:
> | Method | CIFAR10-C | CIFAR100-C | _Avg_ |
> |---|:---:|:---:|:---:|
> | Source | 42.3 ± 1.1 | 66.6 ± 0.1 | 54.43 |
> | **IABN*** | 27.1 ± 0.4 | 60.8 ± 0.2 | 43.98 |
> | IABN | 24.6 ± 0.6 | 54.5 ± 0.1 | 39.53 |
> | __IABN*+PBRS__ | 23.2 ± 0.4 | 55.3 ± 0.1 | 39.26 |
> | IABN+PBRS | **20.1 ± 0.5** | **46.4 ± 0.0** | **33.24** |
>
>
> We note that IABN* still shows a significant reduction of errors under the CIFAR10-C and CIFAR100-C datasets compared with BN (Source). We interpret this as the normalization correction in IABN is valid to some extent without re-training of the model. We notice that IABN* outperforms the baselines in CIFAR10-C with 27.1% error, while the second best, LAME, showed 36.2% error. In addition, IABN* also shows improvement combined with PBRS (IABN*+PBRS). This result suggests that IABN can still be used without re-training the model.
>
> We thank you for the suggestion, and we included these new results in the revised supplementary file (Section C.3 and Table 19), which we believe would be an interesting investigation and discussion about the possibility of skipping retraining with IABN.
>
> ---
> **Q4. For the real-world datasets, what is the source accuracy on the source data? This is important to report for measuring adaptation so that the severity of the shift and how much it is mitigated can be gauged. As presented, it is hard to know if HARTH and ExtraSensory are hard problems, and if the source model is not already doing quite well.**
>
> Thank you for your suggestion. We calculated the source-domain error rates of KITTI, HARTH, and ExtraSensory compared with the target domain(s).
>
> | KITTI | _Src domain_ | Rain | _Avg_ |
> |---|:---:|:---:|:---:|
> | Source | **7.4 ± 1.0** | 12.3 ± 2.3 | 9.9 |
>
> | HARTH | _Src domain_ | S008 | S018 | S019 | S021 | S022 | S028 | S029 | _Avg_ |
> |---|:---:|:---:|:---:|:---:|:---:|:---:|:---:|:---:|:---:|
> | Source | **11.7 ± 0.7** | 86.2 ± 1.3 | 44.7 ± 2.1 | 50.4 ± 9.5 | 74.8 ± 3.8 | 72.0 ± 2.6 | 53.0 ± 24.0 | 57.0 ± 16.7 | 56.2 |
>
> | ExtraSensory | _Src domain_ | 4FC | 598 | 619 | 797 | A5C | C48 | D7D | _Avg_ |
> |---|:---:|:---:|:---:|:---:|:---:|:---:|:---:|:---:|:---:|
> | Source | **8.3 ± 0.7** | 34.6 ± 2.5 | 40.1 ± 0.7 | 63.8 ± 5.7 | 45.3 ± 2.4 | 64.6 ± 3.7 | 39.6 ± 6.8 | 63.0 ± 3.9 | 44.9 |
>
> As shown, there exists a clear gap between the source and the target domains, which demonstrates the severity of the shift. We included this result in explaining the details of the datasets in the revised supplementary material (Section C.2.2, Table 16, Table 17, and Table 18).

---

> ### Author Response · Authors · 2022-08-02
> **Official response to Reviewer 5mFV (2/4)**
>
> ---
> **Q1.  ImageNet-C in the i.i.d. and non-i.i.d. settings with ResNet-18 (and ideally ResNet-50). For the non-i.i.d. setting, consider the same proposed Dirichlet ordering, or simply ordering the data by classes.**
>
> Thank you for your suggestion. We tested NOTE and the baselines on ImageNet-C that contains 15 types of corruptions for 50,000 test samples, which is a total of 750,000 test samples. We adopt the most severe level of corruption (level-5) following previous studies. We used ResNet18 and simply ordered the data by classes. We kept the hyperparameters of NOTE the same as in the other experiments in our paper.
>
> We specify the error rate on classifying among 1,000 categories for 15 corruption types and also report the averaged error rates. Bold type indicates those of the lowest classification error. The lower, the better.:
>
> (1) Classification error (%) for ImageNet-C with temporally correlated test streams (non-i.i.d):
> | | Gauss | Shot | Impulse | Defocus | Glass | Motion | Zoom | Snow | Frost | Fog | Bright | Contrast | Elastic | Pixelate | JPEG | _Avg_ |
> |---|:---:|:---:|:---:|:---:|:---:|:---:|:---:|:---:|:---:|:---:|:-----:|:---:|:---:|:---:|:---:|:---:|
> | Source | 98.4 | 97.7 | 98.4 | 90.6 | 92.5 | 89.9 | 81.8 | 89.5 | 85.0 | 86.3 | 51.1 | 97.2 | 85.3 | 76.9 | 71.7 | 86.2 |
> | BN Stats | 98.3 | 98.1 | 98.4 | 98.7 | 98.8 | 97.8 | 96.6 | 96.2 | 96.0 | 95.1 | 93.1 | 98.6 | 96.3 | 95.6 | 96.1 | 96.9 |
> | ONDA | 95.1 | 94.7 | 95.0 | 96.2 | 96.1 | 92.5 | 87.2 | 87.4 | 87.8 | 82.7 | 71.0 | 96.4 | 84.9 | 81.7 | 86.1 | 89.0 |
> | PL | 99.3 | 99.3 | 99.4 | 99.5 | 99.4 | 99.5 | 98.8 | 99.1 | 99.1 | 98.1 | 97.3 | 99.7 | 98.4 | 98.5 | 98.5 | 98.9 |
> | TENT | 98.3 | 98.1 | 98.4 | 98.7 | 98.8 | 97.8 | 96.6 | 96.2 | 96.0 | 95.1 | 93.2 | 98.6 | 96.3 | 95.6 | 96.1 | 96.9 |
> | LAME | 98.1 | 97.1 | 98.0 | **87.8** | **90.9** | 87.1 | **78.4** | 87.1 | 80.2 | 81.5 | **39.8** | 96.4 | 82.5 | 70.7 | **64.8** | 82.7 |
> | CoTTA | 98.1 | 98.1 | 98.3 | 98.7 | 98.8 | 97.7 | 96.8 | 96.6 | 96.2 | 95.3 | 93.5 | 98.8 | 96.5 | 95.6 | 96.3 | 97.0 |
> | NOTE | **94.6** | **93.7** | **94.5** | 91.3 | 91.1 | **83.3** | 79.1 | **79.3** | **79.0** | **66.9** | 48.4 | **94.2** | **76.3** | **61.8** | 76.6 | **80.7** |
>
> (2) Classification error (%) for ImageNet-C with uniformly distributed test streams (i.i.d):
> | | Gauss | Shot | Impulse | Defocus | Glass | Motion | Zoom | Snow | Frost | Fog | Bright | Contrast | Elastic | Pixelate | JPEG | _Avg_ |
> |---|:---:|:---:|:---:|:---:|:---:|:---:|:---:|:---:|:---:|:---:|:-----:|:---:|:---:|:---:|:---:|:---:|
> | Source | 98.4 | 97.7 | 98.4 | 90.6 | 92.5 | 89.9 | 81.8 | 89.5 | 85.0 | 86.3 | 51.1 | 97.2 | 85.3 | 76.9 | 71.7 | 86.2 |
> | BN Stats | 89.4 | 88.6 | 89.3 | 90.8 | 89.9 | 81.3 | 69.9 | 72.6 | 73.8 | 62.6 | 44.1 | 92.1 | 64.6 | 60.4 | 70.7 | 76.0 |
> | ONDA | 89.2 | 88.2 | 89.0 | 91.0 | 90.0 | 81.5 | 69.6 | 72.6 | 73.8 | 62.6 | 44.0 | 92.1 | 64.3 | 60.1 | 70.1 | 75.9 |
> | PL | 88.5 | 86.7 | 89.6 | 92.2 | 92.1 | 82.2 | 64.4 | 69.8 | 79.4 | **55.9** | 44.0 | 97.6 | **57.8** | 52.6 | **60.4** | 74.2 |
> | TENT | 92.9 | 90.8 | 92.8 | 95.4 | 94.5 | 88.2 | 74.9 | 74.1 | 83.6 | 56.9 | 44.9 | 98.2 | 58.5 | **52.5** | 64.1 | 77.5 |
> | LAME | 98.6 | 97.8 | 98.6 | 90.7 | 92.7 | 89.9 | 81.9 | 89.9 | 85.0 | 86.5 | 51.1 | 97.3 | 85.6 | 77.1 | 71.8 | 86.3 |
> | CoTTA | **85.6** | **84.5** | **85.5** | 87.6 | 86.3 | 74.6 | **64.1** | 67.9 | 69.8 | 56.1 | **42.7** | 89.0 | 60.0 | 54.3 | 64.8 | 71.5 |
> | NOTE | 87.7 | 85.8 | 87.4 | **83.2** | **83.3** | **73.6** | 65.5 | **65.0** | **68.5** | 58.0 | 43.6 | **75.9** | 61.2 | 54.1 | 62.9 | **70.4** |
>
>
>
> As shown, this experiment with ImageNet-C shows consistent outcomes with the experiments in our paper; (1) NOTE outperforms the baselines under temporally-correlated scenarios while most of the baselines fail to surpass the Source method. (2) NOTE also shows comparable performance (in fact, slightly better than) to the state-of-the-art baseline (CoTTA) in the i.i.d. scenario.
>
> We appreciate your suggestion, and we believe this result would significantly improve the quality of the paper. We will include more comprehensive results, e.g., error bars with multiple runs, in our final manuscript.

---

> ### Author Response · Authors · 2022-08-02
> **Official response to Reviewer 5mFV (1/4)**
>
> We sincerely appreciate your effort and time in offering us thoughtful comments. We respond to each of your questions and concerns one-by-one in what follows. We also ask you to kindly refer to the *common response* we have posted together.
>
> ---
> **Comment1. NOTE has to alter training by replacing BN with IABN, and therefore cannot be applied to off-the-shelf models without re-training. The test-time adaptation setting is in part motivated by not having access to the source/training data, so this makes NOTE less of a good fit for this particular assumption.**
>
> We agree that IABN requires retraining off-the-shelf models. However, we emphasize that the limitation is not severe due to the following reasons:
> (1) Source data is usually accessible: Practitioners typically have their own data to train the model, and thus they can use IABN instead of BN when training from scratch. In addition, popular off-the-shelf models (such as ResNet) are usually trained from public datasets. Retraining the off-the-shelf models would be trivial, given the amount of testing that follows after the deployment.
> (2) As we answer in your Q3 (please see below), simply replacing BN with IABN without re-training still shows the effectiveness of IABN under distributional shifts.
> (3) The other technical component, PBRS, does not require re-training of the model and has similar performance gain as IABN when used solely.
>
> Moreover, NOTE has advantages over the baselines. NOTE requires only a “single” instance for inference which is imperative in real-time applications such as autonomous driving, while most baselines (BN Stats, PL, TENT, LAME, and CoTTA) have to wait for a “batch” for inference. In addition, NOTE needs a single forward for inferring each sample unlike CoTTA that requires 32 forward passes for each sample due to augmentation.
>
>
> ---
> **Comment2. How NOTE updates is not fully specified. While Sec. 3.2 explains that inputs are sampled from the reservoir and batched with test inputs, it does not detail the proportion of test inputs to reservoir inputs and the number of gradient steps among other such considerations.**
>
>
> In our original submission, we explained some of the details of PBRS such as memory size, update frequency, and training epochs in Section 4, line 188-191:
> > We assume the model pre-trained with source data is available for TTA. In NOTE, we replaced BN with IABN during training. We set the test batch size as 64 and epoch as one for adaptation, which is the most common setting among the baselines [32, 4, 40]. Similarly, we set the memory size as 64 and adapt the model every 64 samples in NOTE to ensure a fair memory constraint.
>
> We agree that our original submission lacks some necessary details of PBRS, and thus we added further details on PBRS including Q2 and Q5 in the revised draft.
>
> ---
> **Comment3. The choice of datasets is not entirely justified. HARTH and ExtraSensory are indeed temporal, but at the same time they are unfamiliar for the purpose of evaluating adaptation. Are these datasets subject to shift across the given sources and targets? It would be reassuring to measure source accuracy on source and target to evidence drops and thus the presence of shift. The architectures for these datasets are likewise not standard, which would be fine, except the text does not explain how to they incorporate IABN (though this presumably follows the convolution layers).**
>
> Domain shifts in human activity recognition via sensors are indeed a well-known problem since the advent of smartphones and the ubiquitous computing paradigm. The primary cause of the domain shift is the behavioral and environmental differences between users [r1]; an elderly’s jogging might be confused with a young’s walking. In addition, the placement of mobile devices varies according to the type of device (smartphone vs. smartwatch) and user's preference (hand vs. pocket). While there exists dozens of public human activity recognition datasets, HARTH and ExtraSensory datasets are collected in a free-living environment from tens of users that naturally entails domain shift, and thus we call them “real-distribution” datasets.
>
> Following your suggestion, we report the source accuracy on both source and target in Q4, which shows the domain gap and the difficulty of the problem.
>
> Regarding the architecture, BN (or IABN) layers are followed by the convolutional layer. We revised the draft to add this information (Section 4.2). Thank you for pointing this out.
>
>
> [r1] Stisen, Allan, et al. "Smart devices are different: Assessing and mitigating mobile sensing heterogeneities for activity recognition." Proceedings of the 13th ACM conference on embedded networked sensor systems. 2015.

---

### Official Review · Reviewer_GmfJ · 2022-07-27

**Rating:** 7
**Confidence:** 3
**Soundness:** 3 good
**Presentation:** 3 good
**Contribution:** 3 good

**Summary:**

The authors present a method named NOTE to address the online adaptation for non-iid data streams without additional annotation. It has two key components: instance-aware batch norm (IABN) and prediction-balanced reservoir sampling (PBRS). The former computes the difference between learned knowledge and the current observation (i.e., the new instances from the online data stream). The latter aims to avoid overfitting towards the non-iid streams by mimicking the iid samples with a simulated memory. They evaluate the method with sota TTA baselines on a few datasets (mainly for computer vision) and show that NOTE is effective.

**Questions:**

N/A

**Limitations:**

Yes

**Strengths And Weaknesses:**

***Strengths***


- The paper is comprehensive and has a nice presentation. It introduces the background knowledge very well and the proposed components have clear motivation.

- The IABN and PBRS are both novel and show effectiveness over other baseline methods.

- This paper studies a very important problem of test-time adaptation without supervision. It can be used for many real-world applications.

***Weakness***
- In the experiments, the authors should use more tasks such as some NLP data and more popular base models like Transformer-based ones for both CV and NLP. Otherwise, it's hard to measure how the proposed methods really work in modern AI systems.

---

> ### Author Response · Authors · 2022-08-02
> **Official response to Reviewer GmfJ**
>
> We sincerely appreciate your time and effort in providing us with positive comments. We respond to your question in what follows. We also ask you to kindly refer to the *common response* we have posted together.
>
> ---
> **Comment1. In the experiments, the authors should use more tasks such as some NLP data and more popular base models like Transformer-based ones for both CV and NLP. Otherwise, it's hard to measure how the proposed methods really work in modern AI systems.**
>
> We would like to highlight that all the SOTA baselines (BN Stats, ONDA, PL, TENT, CoTTA) except for LAME, are not applicable to Transformer-based models. Similarly, NOTE is not applicable to models without BN layers. In our original submission, we clarified and discussed this limitation in Section 6:
> > Similar to existing TTA algorithms [28, 32, 40, 43], **we assume that the backbone networks are equipped with BN layers**, and particularly, we replaced the BN layers with IABN layers. While BN is a widely-used component in deep learning, there exist architectures that do not embed BN layers such as LSTMs [13] and Transformers [38]. Whether naively inserting BN or IABN would be sufficient for applying these TTA methods is still in question. A recent study evidenced that BN is advantageous in Vision Transformers [44], showing potential room to apply our idea to architectures without BN layers. However, more in-depth studies are necessary to identify the actual applicability of BN (or IABN) to those architectures.
>
> Nevertheless, the other technical component of our method, PBRS, can be implemented on transformers. To investigate the impact of PBRS on Transformer-based models for NLP tasks, we adopted the **BERT** [r1] tiny model. Specifically, PBRS was utilized to update the trainable parameters in the Layer Normalization layers of the model. We conducted test-time adaptation on common text sentiment analysis datasets, following previous work [r2]; we used  **SST-2** [r3] (movie reviews; source domain) to fine-tune the BERT model and evaluated its test-time adaptation capabilities on **FineFood** [r4] (food reviews; target domain). This setup is valuable for understanding how TTA algorithms perform under the domain gap in text sentiment caused by the difference in topics. During test-time adaptation, we used the Dirichlet distribution for simulating the non-i.i.d. streams, as previously described in our experiment section. The result is presented in the table below:
>
> Table.  Classification error (%) on FineFood with both temporally correlated and uniform test streams. The lower, the better. N/A refers to “not applicable.”
> | | non-i.i.d. | i.i.d. |
> |---|---:|---:|
> | Source | 37.4 ± 0.0 | 37.4 ± 0.0 |
> | BN Stats | N/A | N/A |
> | ONDA | N/A | N/A |
> | PL | N/A | N/A |
> | TENT | N/A | N/A |
> | LAME | 35.7 ± 0.3 | 41.6 ± 0.3 |
> | CoTTA | N/A | N/A |
> | **NOTE (PBRS-only)** | **34.8 ± 2.1** | **34.6 ± 2.5** |
>
> We found that, to some extent, PBRS exhibits its effectiveness. Nevertheless, NOTE cannot fully utilize the synergy between IABN and PBRS regarding balanced statistics updates in this case. While LAME is applicable to models without BN, **LAME’s critical limitation is the performance drop in i.i.d. scenarios**, as shown not only for this particular experiment but also in our main evaluation. The primary reason is, as stated by the authors of LAME, that it “discourages deviations from the predictions of the pre-trained model,'' and thus it “does not noticeably help in i.i.d and class-balanced scenarios.”
>
> In summary, while NOTE shows its effectiveness in **both** non-i.i.d and i.i.d. scenarios, a remaining challenge is to design an algorithm that generalizes to any architecture. We believe the findings and contributions of our work could give valuable insights to future endeavors on this end. We submitted a revised draft to incorporate this discussion.
>
>
> [r1] Devlin, Jacob, et al. "BERT: Pre-training of deep bidirectional transformers for language understanding." arXiv preprint arXiv:1810.04805 2018.
>
> [r2] Moon, Seung Jun, et al. "Masker: Masked keyword regularization for reliable text classification." AAAI 2021.
>
> [r3] Socher, Richard, et al. "Recursive deep models for semantic compositionality over a sentiment treebank." EMNLP 2013.
>
> [r4] McAuley, et al. "From amateurs to connoisseurs: modeling the evolution of user expertise through online reviews." WWW 2013.

---

### Author Response · Authors · 2022-08-02
**Common response to all the reviewers**

Dear reviewers,

We appreciate all of you for your **positive reviews**, and highlighting the strengths of our work:
- **GmfJ**: very important problem, comprehensive, nice presentations, well-explained background, clear motivation, novel methodology, and effectiveness over baselines.
- **5mFV**: well-motivated, clear illustration, sound experimental design on standard datasets, strong results, strong baselines, and consistency of hyperparameters.
- **bg8w**: addressing a practical concern in TTA, and extensive experiments.
- **AWKz**: timely and important topic, pointing out an important issue of existing studies, and validation with synthetic/realistic benchmarks.

We also sincerely thank reviewers for their **constructive** comments to improve our manuscript. We have addressed all the questions raised by reviewers with new experiments during this rebuttal period. We summarize how we addressed the **reviewers’ main questions** as follows:

1. **5mFV**: We evaluated NOTE and the baselines on *ImageNet-C*, showing the effectiveness of NOTE on a large-scale dataset.
2. **5mFV**: We evaluated the performance in the source domain for HARTH and ExtraSensory datasets, demonstrating the difficulty of the problem.
3. **5mFV**: We evaluated the effect of IABN without re-training the source model.
4. **GmfJ** and **bg8w**: We discussed and evaluated the applicability of NOTE and the baselines to a Transformer-based model (BERT) and NLP tasks.
5. **5mFV** and **AWKz**: We clarified missing details of our methodology. We also detailed the figure describing our methodology (Figure 3).
6. **bg8w**: We included an analysis of real-time accuracy change.

We submitted our **revised draft and supplementary file** that addressed individual concerns. We marked changed parts with blue fonts. In summary, we made the following changes:
1. We changed Figure 3 (method overview) for a better illustration of IABN.
2. We specified the output of IABN.
3. We elaborated on the hyperparameters of our method.
4. We detailed the process of adaptation and inference of our method.
5. For tables, we marked degraded performance after adaptation in red fonts.
6. We renamed “real distributions with domains shift” to “real-distributions with domain shift” to avoid confusion.
7. We explained why LAME (one of the baselines) does not work on i.i.d. scenarios.
8. We specified the model architectures for the HARTH and ExtraSensory models.
9. We revised our discussion about models without BN layers.
10. We included an evaluation of the real-time performance changes for the real-distribution datasets.
11. We included an evaluation of the error rates on the source domain for the real-distribution datasets.
12. We included an evaluation of replacing BN with IABN during test time.
13. To meet the 9-page limit, we temporarily moved the related work section to the supplementary file.


Thank you for your consideration,

Authors

---

### Author Response · Authors · 2022-08-07
**A gentle reminder**

Dear reviewers,

Thank you for your time and efforts again in reviewing our paper.

We kindly remind that the discussion period will end soon (in a few days).

We believe that we sincerely and successfully address your comments, with the results of the supporting experiments.

If you have any further concerns or questions, please do not hesitate to let us know.

Thank you very much!

Authors

---

### Meta-Review · Area_Chair_VJH1 · 2022-08-28

**Recommendation:** Accept
**Confidence:** Certain

**Metareview:**

The paper proposed two test-time adaptation methods a) instance-aware batch normalization and b) prediction-balanced reservoir sampling and used these to show that the proposed method is better in the non-iid setting.

The reviewers found this to be an important problem the experiments generally convincing. Reviewers objected the choice of dataset (not commonly used to evaluate adaptation) and baseline models (not state of the art models) and the effect size. In the end all reviewers found the results strong enough and voted to accept.

**Award:**

No

---

### Decision · Program_Chairs · 2022-09-14

Accept